# Transformation of amorphous calcium phosphate to bone-like apatite

Antiope Lotsari [1,2], Anand K. Rajasekharan [1], Mats Halvarsson [2] & Martin Andersson [1]

Mineralisation of calcium phosphates in bone has been proposed to proceed via an initial amorphous precursor phase which transforms into nanocrystalline, carbonated hydroxyapatite. While calcium phosphates have been under intense investigation, the exact steps during the crystallisation of spherical amorphous particles to platelet-like bone apatite are unclear. Herein, we demonstrate a detailed transformation mechanism of amorphous calcium phosphate spherical particles to apatite platelet-like crystals, within the confined nanodomains of a bone-inspired nanocomposite. The transformation is initiated under the presence of humidity, where nanocrystalline areas are formed and crystallisation advances via migration of nanometre sized clusters by forming steps at the growth front. We propose that such transformation is a possible crystallisation mechanism and is characteristic of calcium phosphates from a thermodynamic perspective and might be unrelated to the environment. Our observations provide insight into a crucial but unclear stage in bone mineralisation, the origins of the nanostructured, platelet-like bone apatite crystals.

[1] Department of Chemistry and Chemical Engineering, Chalmers University of Technology, SE-412 96 Gothenburg, Sweden. [2] Department of Physics, Chalmers University of Technology, SE-412 96 Gothenburg, Sweden. Correspondence and requests for materials should be addressed to M.A. (email: martin.andersson@chalmers.se)

The basic composite structure of bone is observed at the nanoscale where unidirectionally aligned collagen fibrils are densely mineralised with inorganic nanoplatelets of hydroxyapatite (HAp)[1,2]. Apatite in bone has unique geometry and chemical composition; a crystal does not exceed 30–50 nm in length, it maintains ≈2 nm thickness, it has an imperfect platelet geometry and poor crystallinity and it is deficient in calcium and substituted with carbonate[3,4].

There is increasing in vivo and in vitro evidence indicating an amorphous calcium phosphate (ACP) precursor phase, which could possibly be allocated and deposited by intracellular and mineral containing vesicles at the gap zones of the collagen matrix. It is theorised that ACP acts as a transient phase which is easily transferred as a precursor for the apatite growth. For example, ACPs have been observed in continuously growing bones of zebrafish fin, developing mouse calvaria and embryonic chicken long bones. The ACPs observed in these studies were either spherulites or globules ranging between 50–80 nm in size that gradually transforms to apatite with elongated and platelet-like morphology, aligning along the long-axis of the fibril[5–11]. In vitro studies, using collagen as the organic scaffold, have also shown the presence of ACPs at the gap zones of collagen prior to its transformation to apatite[6,9,10].

With the growing evidence for an ACP mediated growth of apatite in vivo and in vitro, there is a lack of clear evidence for the precise crystallisation process, whereby a spherical or irregular amorphous component ($d ≈ 50–80$ nm) continuously crystallises to form a thin platelet-like crystal ($50 × 30 × 2$ nm$^3$)[5,7,9,11]. Many in vitro studies have emphasised the importance of pre-nucleation clusters during ACP-HAp transformation. The first theory was proposed by Posner and Betts who observed the presence of pre-nucleation clusters of $Ca_9(PO_4)_6$ of ~1 nm size (so-called 'Posner's cluster'). They proposed that these clusters not only could be the building block for ACP formation but also play an important role during its transformation to apatite, suggesting that ACP only dissociate into these clusters rather than undergo a complete ionic dissolution. These clusters then slowly re-arrange themselves to form the HAp structure by taking up additional OH and Ca ions[12]. More recently, a cryo-Transmission Electron Microscopy (cryo-TEM) study by Dey et al. demonstrated ~1 nm clusters during the earliest stages of CaP nucleation on a Langmuir monolayer of arachidic acid to mimic bioimineralization of CaP in vivo[13]. Pre-nucleation clusters of similar sizes, that consist of calcium triphosphate $[Ca(HPO_4)_3]^{4-}$ ion-associated complexes, have also been suggested to aggregate and form ACP particles by taking up an extra $Ca^{2+}$ ion[14]. It is particularly notable that this proceeds under similar conditions where the ACP to HAp transformation occurs.

The first experimental evidence of the presence of pre-nucleation clusters during the growth of apatite crystals was presented by Onuma et al; by using in situ Atomic Force Microscopy (AFM), they observed a step growth along the $[1\bar{1}00]$ direction[15]. They proposed a cluster growth model incorporating the Posner's cluster in the HAp growth process[16] and theorised that the growth proceeds along $[1\bar{1}00]$ direction via a combination of two-dimensional (2D) nucleation and step flow[17,18]. A similar study which also utilised in situ AFM, reported a step growth of HAp with similar step sizes during a classical spiral growth (due to dislocations with a screw component) on $(01\bar{1}0)$ HAp surfaces with a simultaneous non-classical growth through particle attachment[19]. In this case, dissolution of the HAp crystal surface takes place before the growth is initiated. CaP clusters matching the Posner's cluster size were also observed at calcite-water interface[20]. The pre-nucleation cluster theories are crucial to fully unravel the detailed transformation of ACP to apatite, since these clusters are proposed to act as the building blocks. However, a

growth mechanism that incorporates a step growth or cluster incorporation from the amorphous phase has not yet been reported, while there is visual evidence of the ACP-apatite transformation.

Considering the challenges in probing this dynamic process in vivo, we propose in this work a bone-mimetic system as a model to observe the ACP-apatite transformation process. We employ high-resolution transmission electron microscopy (HRTEM) in order to elucidate the early stages of ACP crystallisation and how nucleation clusters might play a role in the transformation process. Considering the importance of ACPs in bone apatite mineralisation, a generic mechanism is hypothesised for the in vivo ACP-apatite transformation. The crystallisation route is relevant to both synthetic apatites as well as for other biological materials mineralised with apatite.

## Results

**Mineralisation of ACP within confined polymer domains**. The mineralisation of ACPs and its transformation to apatite nano-crystals was carried out within confined nanodomains of a hex-agonally ordered, polymerised lyotropic liquid crystal (PLLC) matrix. The PLLC is a meso-ordered, soft polymeric matrix, composed of nanoscale micellar fibrils (NMF) of 7–10 nm in diameter, organised in a hexagonal lattice at the sub-micron scale. This type of liquid crystalline arrangement is obtained via the molecular self-assembly of di-acrylate modified, non-ionic, amphiphilic block co-polymers, and aqueous calcium phosphate precursors (see experimental section in ref [21]). Our previous works have demonstrated that the NMFs mimic the nanostructure of bone in two important ways; (i) the NMFs possess a long-range order with confined nanoscale domains or zones (≈10–15 nm) which are similar to the confined domains (≈40 nm gap zones) in collagen and (ii) apatite can be selectively mineralised in situ, within the nanodomains between NMFs, whereby the formation proceeds via an ACP phase and grows into carbonated HAp with bone-like structure and chemistry[21–24]. Except for the confinement effects of the NMF, there is no chemical or electrostatic interactions between the polymer matrix and the calcium phosphate (CaP) particles, which aids in studying the mineralisation and crystallisation process independently. The mineralisation is initiated within the PLLC by increasing pH within the aqueous domains; thereby, precipitating the ACP nanoparticles to form a PLLC-ACP composite material (see Methods). The microstructural evolution of CaP minerals within the PLLC is similar to bone with respect to the initial and final structure and morphology; the first formed particles are spherical ACPs in the size range of 50–100 nm in diameter (as shown in the Bright Field (BF) TEM image of Fig. 1b), which transform to apatite platelets via an aging process. Aging refers to the storage of the PLLC-ACP composite samples in a humid environment at 37 °C for a defined time period (see Methods). Although the size of particles is significantly larger than the aqueous domains located between the NMFs, the fibrils deform and accommodate the growing ACP nanospheres (Fig. 1a). It is important to note that the size of ACPs seen in the present system represents the ACP aggregations observed during bone formation by Mahamid et al[7]. In their final morphology the apatite crystals are elongated along the c-axis and exhibit an irregularly shaped platelet morphology.

**TEM characterisation of CaP particles within the composites**. Following mineralisation, the PLLC-ACP composite was aged at 37 °C for a maximum of 3 weeks in a humid chamber after which all the ACP particles transformed to apatite. Prior to imaging, the composite was manually ground in presence of ethanol to obtain

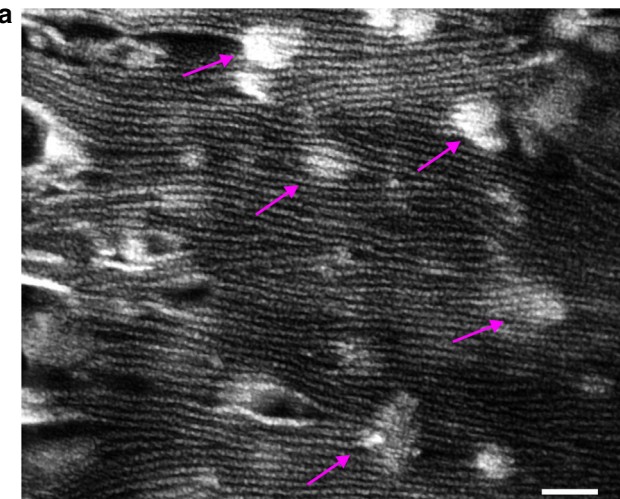

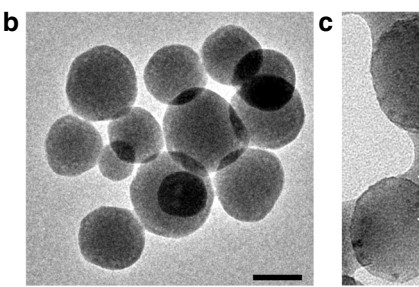
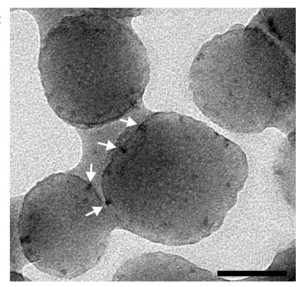

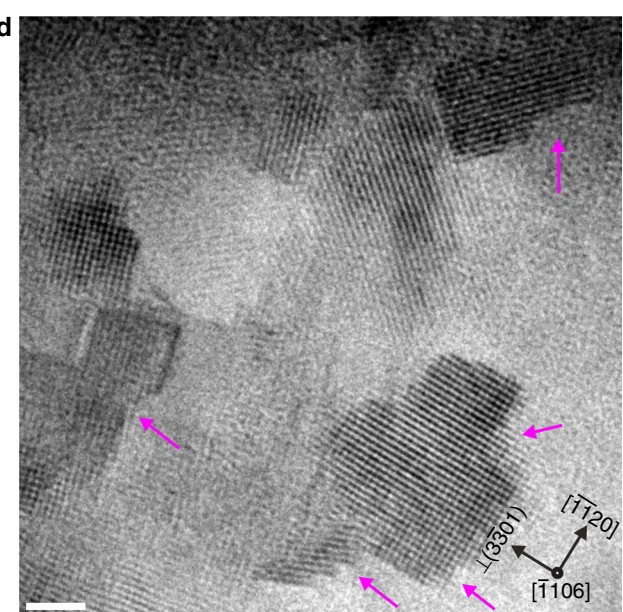

**Fig. 1** Morphology of the composite. **a** SEM image showing the formed ACP particles between the confined domains of the NMFs within the PLLC matrix (arrows). The NMFs appear to be distorted in order to accommodate the large ACP particles. Scale bar, 100 nm. **b** BF TEM image of freshly formed ACP particles. Scale bar, 50 nm. **c** BF TEM image of particles that deviate from the spherical morphology as they begin to coalesce. The arrows denote electron dense areas on the particles. Scale bar, 50 nm. **d** HRTEM image of similar areas denoted in **c** that correspond to the first crystalline nucleation sites of the crystalline phase. Arrows denote steps edges parallel to $(3\bar{3}01)$ and $(\bar{1}1\bar{2}0)$ planes. Scale bar, 2 nm.

a dispersion. A few drops of this dispersion were used to prepare the TEM specimen on a C-coated Cu grid (see Methods). Even for the freshly prepared samples (i.e. aging for less than 1 day) there are areas in which particles start to aggregate and deviate from the spherical morphology indicating that the structural transformation has been initiated. The BF TEM image in Fig. 1c, shows such a particle in the middle that appears to be slightly elongated along one direction. The arrows around this particle and the neighbouring ones indicate electron dense (dark) areas. These electron dense areas are present in all the fresh samples and coexist with the initially formed amorphous particles (like the ones in Fig. 1b), signifying that the aging procedure occurs at varying rates throughout the whole sample. At the same time, this difference in growth rate allows the observation of different aging steps in the same sample.

HRTEM was employed to identify the nature of the electron dense areas located at the periphery of the ACP particles. Initially, the dark areas measuring ~5 nm in size were assumed to be aggregations of ionic clusters that continuously dissolved from the ACP under the humid conditions of the aging procedure. The clusters were thought to correspond to the first crystal units that aid the transformation to apatite via a rearrangement of CaP building blocks, similar to those observed by Habraken et al., since we observed similar arrangements in our samples (Supplementary Figure 1)[14]. However, the HRTEM observations gave a slightly different result. These areas do in fact correspond to the initial crystallisation points as shown in the HRTEM image of Fig. 1d, but instead of dissolved material formed around (outside) the particles, they form from the amorphous matrix on the surface of the ACP particles. There are two main characteristics of these nanocrystallites: (i) they always form first at the edges of the amorphous particles indicating that the transformation to a crystalline structure starts from the surface towards the inside and (ii) they exhibit a stepped morphology (magenta arrows in Fig. 1d). Along with the steps, a common feature observed in the majority of crystallites was the presence of the $[11\bar{2}0]$ direction either as in plane or as the viewing direction of the crystallite. In the case of the $[11\bar{2}0]$ being an in-plane direction, the measurements regarding the step height and width have been separated to those parallel and perpendicular to this direction. The dimensions of all steps observed from multiple crystallites are presented in Supplementary Figure 2.

Along the $[11\bar{2}0]$ direction, the steps are crystallographic and correspond to the $d$-spacing of the $\{01\bar{1}0\}$ planes of HAp (~0.8 nm, ~1.6 nm or a multiple of 0.8 nm). The steps have edges along the $[0001]$ and $[1\bar{1}00]$ directions and match the heights of the steps observed by Onuma et al[15]. This is demonstrated in the HRTEM images of Fig. 2 of a partially crystallised ACP particle. These micrographs have been recorded 2 min apart to minimise beam damage and beam-induced crystal growth. In Fig. 2a, the ACP particle is mainly amorphous with small crystalline domains as denoted by the magenta arrows. These domains are also located along the periphery of the particles which is consistent with our earlier observations in Fig. 1c. After 2 min, the crystallisation front observed in the particle shown in Fig. 2a has advanced further, where steps of $h \approx 0.8$ nm appear on the upper side of the crystallite (yellow arrows in Fig. 2b). The light blue arrows denote a 1.5 nm thick zone at the left-most periphery of the crystallite, which appears to exhibit a diffuse, amorphous-like structure. Images taken at the final time point illustrates a continued crystallite growth within the particle (area marked as 1) where the emergence of a secondary crystalline domain (area marked 2, viewed along $[10\bar{1}1]$ zone axis) is also observed as shown in the HRTEM image of Fig. 2c. In addition to the two large crystallites, a smaller crystalline domain can be observed on the left side of the particle, as indicated by the green arrow.

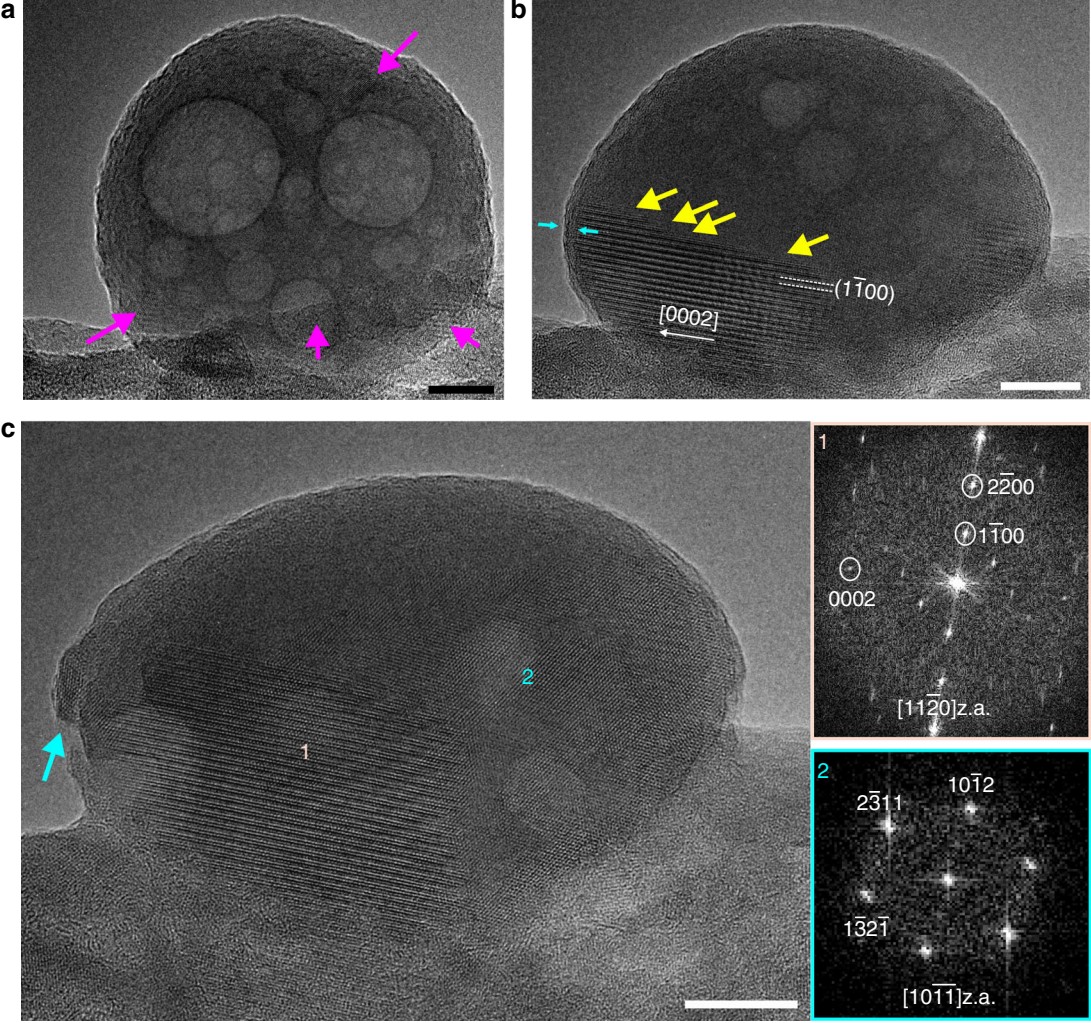

**Fig. 2** Transformation of a partially crystallised ACP particle. **a** HRTEM image of the ACP particle (arrows denote crystallised domains). **b** Same particle after 2 min. The crystalline area on the left-hand side has advanced and the growth proceeds via a layer-by-layer mechanism incorporating steps (yellow arrows). The light blue arrows denote a 1.5 nm thick zone around the edges of the particle. **c** After 4 min, a second crystalline area is observed. The insets correspond to the FFT diffractograms from each area. The cyan arrow denotes a small crystallite. Aging time of the sample was 3 weeks. Scale bar in all images, 10 nm

Occurrence of this small crystallite could explain the polycrystalline nature of the particle surface. The change in shape of the particle as shown in the HRTEM images gives an impression that the particle is being elongated along one direction (which corresponds to the direction of the *c*-axis of the crystalline area depicted in Fig. 2b and almost aligned with the 'width' of the particle). What is interesting with this shape change is that the width remains the same (~62 nm) while the particle's height is decreased from ~57 nm to 47 nm. This indicates that the particle is shrinking normal to the *c*-axis due to the density difference between the amorphous and crystalline phase of CaP. In addition, the particle could grow in thickness along the viewing axis which in this area is the $[11\bar{2}0]$. Since TEM micrographs are 2D projections of 3D structures, it is difficult to interpret which of the two scenarios best explains the morphological change. It is evident though that the final orientation and morphology has not been attained yet since there is a second, fast growing crystalline domain (area 2 in Fig. 2c) with a different *c*-axis direction.

At a final stage the crystal has attained an almost single orientation as can be seen in Supplementary Figure 3b while Supplementary Figure 3a shows the HRTEM image of Fig. 3c. The images have been superimposed on the lattice strain maps

obtained by Geometrical Phase Analysis (GPA)[25,26]. In Supplementary Figure 3a the yellow arrow points to a small crystallite adjacent to the particle that follows the orientation of the crystalline region of area 1.

**Investigation of the presence of octacalcium phosphate.** It should be emphasised that no intermediate CaP phases have been observed during the crystallisation. This, however, cannot confirm the absence of any metastable phase prior to the formation of initial crystalline domains at the edges of the ACP particle. Although octacalcium phosphate (OCP) has been widely proposed to act as a transient metastable phase in mineralisation, its presence is very difficult to identify with TEM due to its high similarity to apatite[14,27]. The two CaP phases differ only in the interplanar spacing of the (100) OCP/$(10\bar{1}0)$ apatite planes. In our observations, the material is being transformed from amorphous to partially crystallised to fully crystallised and even in the initial crystalline areas, the unique to apatite $(10\bar{1}0)$ planes are being resolved, indicating the absence of OCP. To further investigate the possible OCP formation before apatite, we performed X-ray diffraction (XRD) measurements on freshly made composites (unaged) and monitored them up to 5 days of aging

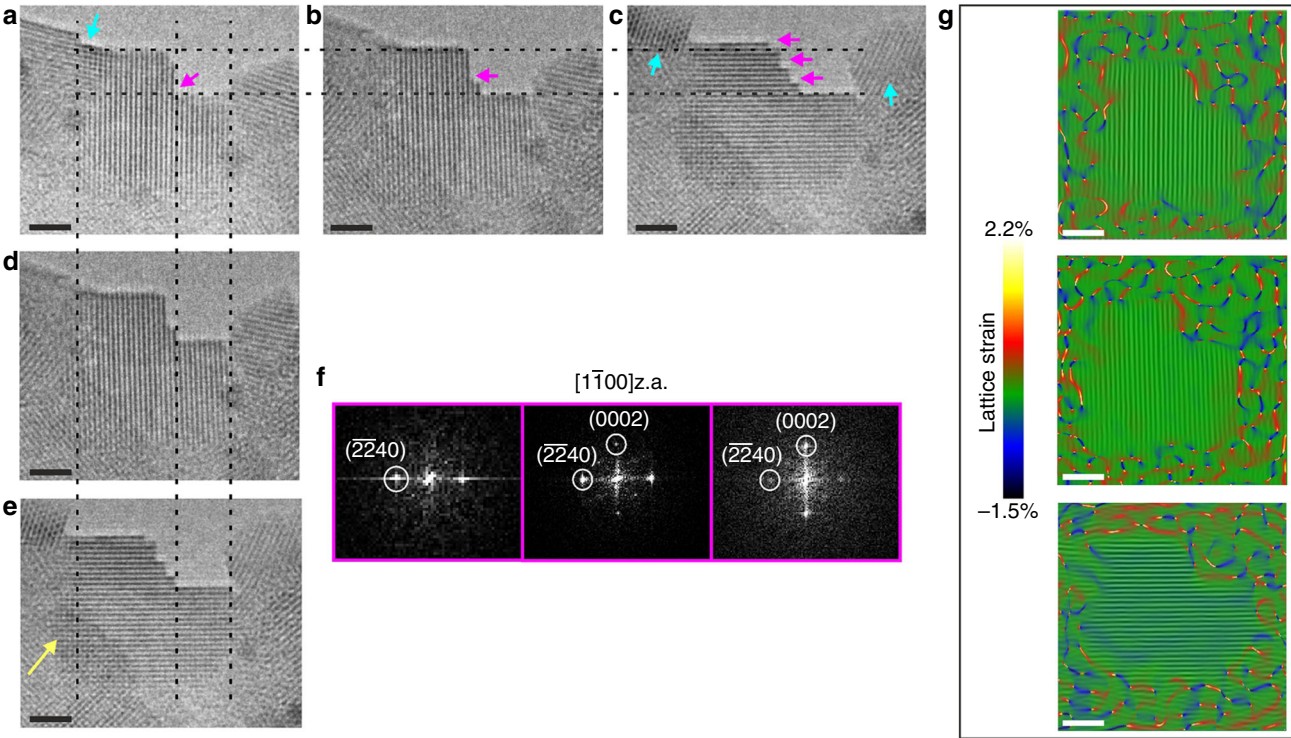

**Fig. 3** Step formation at the growth front. **a** HRTEM image of a small crystallite viewed along the [1$\bar{1}$00].z.a. exhibiting a step (magenta arrow). The edges of the crystallite are diffused. **b**, **d** Same crystallite (placed inline and underneath (**a**)) in which an extra step has been formed. **c**, **e** Three steps have been formed. The dashed lines act as a guide for the eye, taking as reference the initial crystallite's step height and width differences. A ~1–2 ML height difference is visible as well as an increase in the width (denoted by the yellow arrow). Cyan arrows denote the adjacent crystallites that have grown as well. **f** FFT micrographs, from right to left, obtained from the nanocrystals shown in (**a**, **b** and **c**), respectively. **g** Bragg filtered images obtained from the respective HRTEM micrographs in (**a**, **b** and **c**) using g = $\overline{2}\overline{2}40$ and g = 0002 with their corresponding GPA lattice strain maps superimposed, making the boundaries of the crystallite more apparent. Aging time of the sample was 3 weeks. Scale bar in all images, 2 nm

in humid environment at 37 °C. As can be seen in the XRD diffractograms in Supplementary Figure 4, the strongest peaks of OCP, i.e. (100) and (010) at 2 Theta 4.747° and 9.744°, respectively, are absent thus confirming the TEM observations that ACP transforms directly to apatite.

**HRTEM observation of stepped growth of apatite crystallites**. In order to clearly depict the step growth of the crystal, HRTEM images were taken at a different polycrystalline region of the prepared TEM sample. The small crystallite shown in Fig. 3a is located at the crystallisation front of a large sample area (~1 μm), where particles have formed around the adjacent NMFs. Images were acquired in sequence (~2 min apart). In the HRTEM image of Fig. 3a, the edges of the crystallite are not well defined; they appear diffuse and not atomically flat. The crystallite exhibits only one step of 1.9 nm height, which is denoted by the magenta arrow. The dashed lines in Fig. 3a–c, are positioned at the lower and upper surface of the initial step to act as a guide for any morphological change. At the second time point (Fig. 3b), a new step (2.4 nm × 0.48 nm) has formed (magenta arrow) and the edges of the crystallite become much clearer and well defined. At the final time point (Fig. 3c), multiple steps have formed on the ($\overline{2}\overline{2}40$) edge of the initial step. The important feature to note here is the increase in height (as denoted from the upper dashed line) by 0.4 nm meaning that the crystallite has grown along the c-axis and its edges now appear atomically flat. The abiding crystals (cyan arrows) have also grown around 1–2 nm.

The steps that are formed have edges along the [0001] and [11$\bar{2}$0] directions and their direction becomes apparent by

comparing the images vertically (Fig. 3a, d, e), where the dashed lines are now drawn vertically as a guide for the step direction where the width of the lower edge of the first step is taken as a reference. It becomes more evident that the steps are advancing towards the [$\bar{1}\bar{1}20$] direction, which is well depicted in Fig. 3e. This reinforces the conclusion that all the steps follow the same direction. Parallel to the observed increase in height, it appears that the crystallite has also grown along the [$\bar{1}\bar{1}20$] direction. Although the left side of the crystallite is not clearly defined in Fig. 3a, the dashed line has been drawn where the crystallite appears to stop overlapping with the neighbouring one on the left, indicating the termination of the crystallite (area denoted by the cyan arrow). From Fig. 3a, to Fig. 3e, there is an increase in the width of the crystal by 1–1.4 nm together with a possible change in thickness, since the crystal seems to overlap with the crystallite denoted by the yellow arrow in Fig. 3e. The increase of the crystal size as well as the boundaries that define the crystal are illustrated well in Fig. 3g, which depict the GPA lattice strain maps superimposed on the Bragg filtered images obtained from the HRTEM micrographs. In the fast Fourier transformed (FFT) micrographs depicted in Fig. 3f (corresponding to HRTEM images of Fig. 3a–c), the spatial frequencies, used for the lattice strain maps, are indicated by white circles. In addition, in Fig. 3c, the (0002) planes are resolved while the ($\overline{2}\overline{2}40$) planes are less intense, as shown in the third FFT micrograph in Fig 3f. This change could indicate thickness increase, however this is not clear, since such a change could also result from rotation of the whole specimen area (there is an increased sensitivity in the stability due to the presence of the polymer fibrils nearby that are

unstable under the electron beam). It is worth noting here that when we measure the stepped area (confined by the bottom horizontal and the left perpendicular dashed lines) it remains practically the same (~10.5 nm$^2$), assuming constant crystal thickness. This indicates that even if there is no thickness change to this point, crystal growth occurs in width.

**From ACP to apatite: a proposed growth mechanism.** According to the TEM observations, the following mechanisms, incorporating steps, for the transformation of apatite crystallites in our bone-mimetic system could occur. The first is that the forming apatite crystals exist in a highly transient CaP phase consisting multiple crystalline domains along various orientations that re-arrange themselves via the migration of ion clusters through a step flow. This mechanism proceeds toward the faster growing facets and/or to attain the platelet-like morphology. The aforementioned constant surface area in the stepped part of the crystallite strongly suggests material migration in the form of clusters towards the main growth direction of apatite crystals (c-axis). Although such a mechanism could take place by re-orientation of internal crystallite structure[12], it cannot be the sole mechanism since the big difference in included volumes, from the spherical form to the platelet-like morphology strongly indicates that a dissolution mechanism also takes place under the presence of water (average spherical particle: ~60 nm diameter; average apatite platelet size: $4 \times 20 \times 50$ nm$^3$). Partially, this volume change is the result of the difference in density between the two phases, but it is generally accepted that a dissolution–re-precipitation mechanism takes place in apatite transformation, where the material itself acts as a seed for further growth, but also has been shown in situ that such a mechanism occurs simultaneously with stepped spiral growth[19]. Although during our ex situ experiments we cannot have a dissolution process without the presence of a solvent (water), a dissolution–re-precipitation mechanism could be a parallel transformation mechanism in an aqueous environment.

The increase in width of the crystallite depicted in Fig. 3, might be that the crystallite consumes and accumulates material from the neighbouring crystallites, similar to what happens in situ. The step heights, especially in Fig. 2 correspond well to what has been reported in the literature for the $\{01\bar{1}0\}$ steps and the size of the Posner's clusters[12]. While other step dimension (normal to $\{01\bar{1}0\}$) observed in this work have not been reported elsewhere, it is suggested that the step volume could also correspond to the volume of single Posner's cluster. The sometimes long $\{01\bar{1}0\}$terraces between the steps could indicate that the growth proceeds through layer-by-layer growth with the addition of $\{01\bar{1}0\}$planes. However, it is also evident from Fig. 2b that new layers start to form before the complete growth of the first layer, something that is not so consistent with the classical layer-by-layer growth mechanism in epitaxy. This is an indication that a new cluster is accumulated on top of the previous layer rather than incorporation of a new cluster at the step edge[28]. Similar results have been demonstrated by Onuma et al. in AFM experiments on large synthetic apatite crystals[28]. They observed an accumulation of growth units in the terraces between steps that then diffused towards the step front of the crystal where the growth proceeded via surface diffusion. They further report that when the supersaturation degree of the system is high enough to induce two-dimensional growth, this will first occur between the steps and not at the edges.

Comparing the step sizes in Fig. 1d and Fig. 3 with those reported in the literature there are differences in the size of the steps with the dimensions varying over a wide range, which might be due to the difference in the viewing axis reported in this work.

However, it is possible that the current observations are a projection of the previously reported clusters. At the same time, the step dimensions are similar or within the range of the sizes of pre-nucleation clusters that has been observed in multiple in vitro studies[14,15] when the viewing axis is the same, i.e. along [11$\bar{2}$0].

Based on the above results, we propose a combinatorial growth model for apatite crystallites from ACPs that involve a classical-non-classical mechanism where internal material migration proceeds in the form of clusters through a step flow, accumulation of material from the surrounding environment and lastly, a process equivalent to the classical layer-by-layer growth. Here, the clusters promote crystallisation along the c-axis. After this step, the crystal growth could proceed layer-by-layer with adatom attachment, which either pre-exists in the initial precursor solution or sourced from dissolved ionic clusters during aging. The structural and morphological transformation of ACP to apatite is a complex process. The step-flow migration of material, as observed here, could be part of the transformation process in combination with additional mechanisms such as dissolution and precipitation. In the presence of water, a dissolution–re-precipitation mechanism could possibly occur in coordination with the above steps. The crystal growth mechanism from ACP to apatite follows the thermodynamically driven phase transformation that is governed by the solubility of the CaP phases[29]. For this transformation, a humid environment and ambient conditions are necessary and the orientation and location of the crystals are hindered by the polymeric network's space confinement.

It appears that multiple crystal growth mechanisms are involved during the transformation of ACP to apatite, which might explain the irregular shape of the fully transformed (aged) crystals that are similar to bone apatite[29]. Step traces are evident even in the fully-grown crystals, two of which are shown in the HRTEM images of Fig. 4a, b. Both crystals are viewed along the [11$\bar{2}$0] zone axis (z.a.) and they have an elongated morphology along the c-axis. Magenta arrows denote the steps on the sides of the crystals on the $\{01\bar{1}0\}$ planes. The successive steps, especially in the crystal depicted in Fig. 4b results in a thickness reduction from 4.5 nm to 2.7 nm. A 1.5 nm zone similar to what was observed in Fig. 2b appears to surround the crystal from the sides but not at the apex, where the crystal appears to be bounded in a truncated manner by $\{01\bar{1}1\}$ facets forming ~40° with the (0002) planes. Taking into account the proposed material dissolution, the reason as to why this zone appears diffuse might lie in the fact that the side facets are the dissolution-like fronts. This is in close resemblance to the mottled morphology in the dissolution front in Si-HAp bioceramic implants[30]. Similarly, disordered surface layers have been identified around the crystalline core of calcified biominerals in various natural materials. A study by Wang et. al. suggests that such a surface layer provides an interface through which water molecules are able to orient the apatite crystals in vitro, without the presence of any biological component[31]. In our experiments such a disordered surface layer has been identified at all the aging stages (water induced). These observations trigger the need to re-evaluate the exact role of biological components present in vivo that have been proposed to promote and guide the growth of bone apatite[6,32].

The transformation of ACP to apatite is a process that is common in both synthetic and biological systems. This is expected since apatite is the most thermodynamically stable CaP phase under physiological conditions. In our specific case, the resulting apatite crystals are similar to the apatite found in bone, both in terms of morphology and crystallinity[29].

Observing along the [11$\bar{2}$0] direction, it appears that the facets describing the crystal are well defined; however, this is not the case when the crystals are viewed from the other side facets. An example is shown in Fig. 4c, where the sides of the crystal form

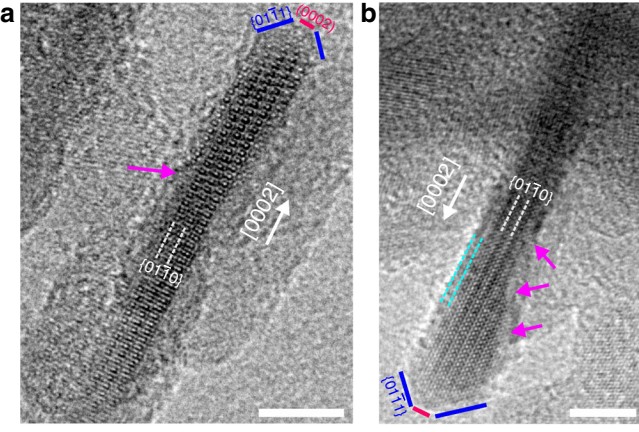

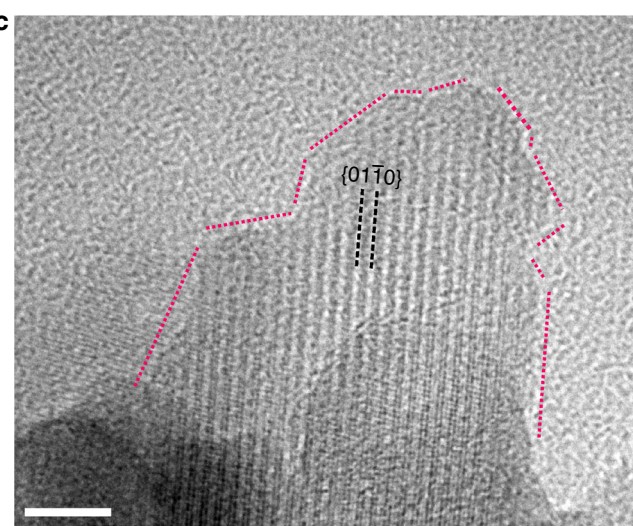

**Fig. 4** Morphology of aged crystals. **a**, **b** HRTEM images of two apatite crystals viewed along the $[\bar{1}\bar{1}20]$z.a. The magenta arrows denote steps with edges along [0001] and $[1\bar{1}00]$directions. The termination of the apatite crystals is truncated and bounded on the one edge by $\{01\bar{1}1\}$facets. Cyan dashed lines in **b** denote a 1.5 nm diffuse area around the crystal. **c** Side view of an apatite crystal with an irregular shape bounded by facets of various orientations. Scale bar in all images, 5 nm

various angles with the $\{01\bar{1}0\}$planes resulting in a distorted crystal making it difficult to construct a definite morphological model to accurately depict the crystal morphology (magenta dashed lines). The deviation from a perfect HAp single crystal morphology is evident and this agrees well with apatite crystals observed in biology, which also exhibit very poor crystallinity and an irregular crystal structure.

In order to unite the different processes discussed in this work, an illustration of the ACP-apatite transformation within the confined domains of the NMFs is shown in Fig. 5. As pH increases, the calcium and phosphate ions within the polymer matrix precipitate to form spherical ACP particles. This is followed by the step growth of nanometre sized domains in which material migrates along their c-axis in the form of clusters. As the crystallisation is initiated the particle attains a transient polycrystalline character starting from the outer part of the particle towards the inside. The faster growing crystalline areas (that grow fast along the c-axis) could consume, incorporate, or force smaller crystallites in their proximity to follow the main orientation, resulting in the initial deviation from a spherical morphology. As the particle starts to follow a single orientation, a step flow of ionic clusters towards the apex of the crystals is

promoting growth along the c-axis. The migration of material from the crystal's sides, results in decrease of thickness and when the growth stops, traces of the steps are evident on the facets that bound the sides of the crystal axis.

Our observations on ACP-apatite transformation correlate well with studies performed on single crystals of synthetic HAp which indicate that the presence of pre-nucleation clusters and the formation of steps of sizes similar to the Posner's cluster is an important step in the ACP-apatite transformation process[15–17].

**Relevance to bone mineralisation**. From a biological perspective, our observations support earlier hypotheses that stabilisation of bone apatite from ACP precursors, post mineral deposition, might purely be a result of thermodynamics with only the aid of water[31,33]. At the same time, the biological components such as collagen, non-collagenous proteins, glycosaminoglycans, intra-, and extracellular vesicles are crucial for the transport, deposition and binding of the ACP components. However, it is possible that the actual transformation of the amorphous precursor to bone apatite and its subsequent growth might not require any chemical interactions between the organic components and the CaP phase. The hypotheses is based on observations from our bone-mimetic model, which is suitable due to its relevance to the natural systems based on the following; (i) the ACP particles formed in our system closely resemble the amorphous granules in size and shape as observed in intracellular vesicles and mineralising collagen fibrils from multiple in vivo studies[7–9], (ii) the final apatite crystal resembles bone apatite in size, crystallinity, composition, and the irregular platelet-like geometry[21,22], (iii) the precipitation of ACP, its growth and transformation to apatite occurs within confined domains of the PLLC matrix, similar to the gap zones seen in bone, and (iv) we have earlier demonstrated the formation of CaP phases within confined domains and at high supersaturations selectively forms ACP particles which transforms to bone-like apatite. It is important to note that the interfibrillar spacing of the PLLC matrix is ~10 nm, where the CaP species nucleate into ACP particles, further transforming into apatite. In collagen, the initial mineral (ACP) deposition is at the 40 nm long gap zones of the collagen fibrils, followed by the growth and maturation of apatite platelets, leading to its penetration into the fibril spaces of 1.5 nm. Even though the final fibril spacing is different between the PLLC and collagen, it must be emphasised that it is the similarity in localisation of the initial mineral deposition, (i.e. ACP and its subsequent transformation) that is the primary basis for comparing the PLLC of this work and collagen systems in vivo. Although, it is possible that the 1.5 nm spaces between collagen fibrils might contribute for a 2 nm thickness of bone apatite, it is notable that the actual thickness of bone apatite crystals falls within a range of 2–6 nm[1]. In this work, the thickness of the apatite platelets formed within the PLLC also falls in a similar range (≈4 nm). Therefore, the PLLC matrix provides a confined domain of a specific size, which is essential to obtain a final apatite particle similar to those observed in bone. Moreover, we have previously shown that the size of the PLLC confinements plays an important role in controlling the type of CaP formed. Confinements exceeding 10 nm results in a mixture of CaP phases while under 10 nm, the CaP particles purely resembled bone-like apatite platelets[23]. Moreover, the confinement also restricts the growth of the crystallites in the range of 30–40 nm in length, while in absence of the PLLC matrix confinements, the crystallites could exceed 200 nm in length[22]. Therefore, it is clear that the 10 nm confinements of our PLLC system is able to select for ACP, constrains the size via elastic deformation and favours the growth of apatite crystallites with thickness similar to bone apatite and with their c-axis aligned along the length of the micellar fibrils.

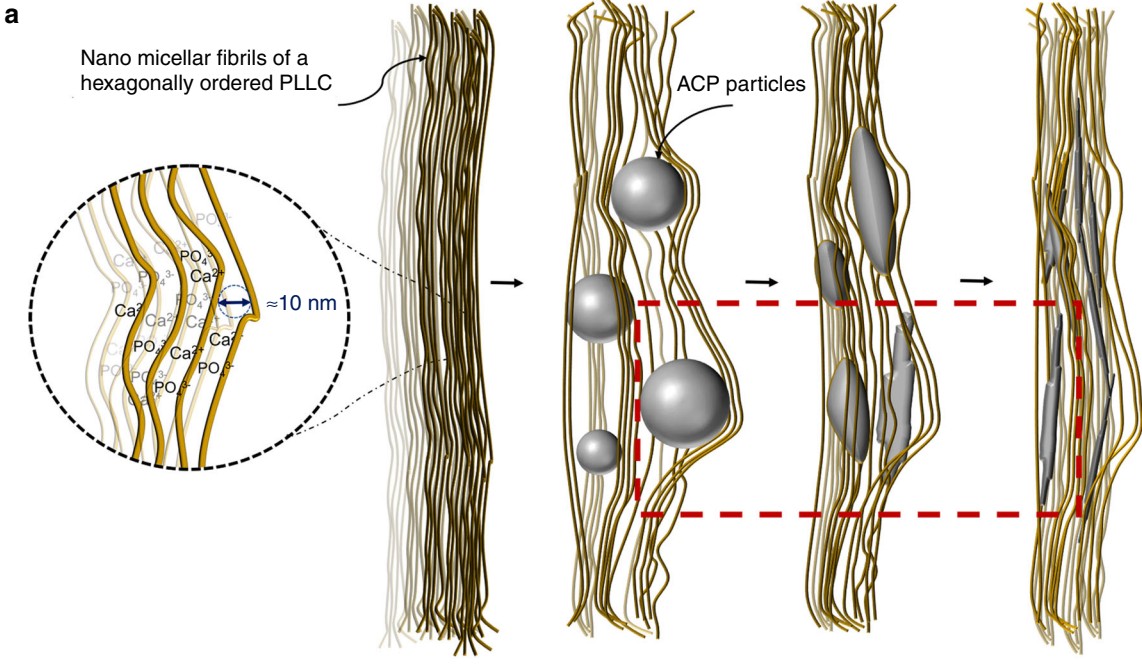

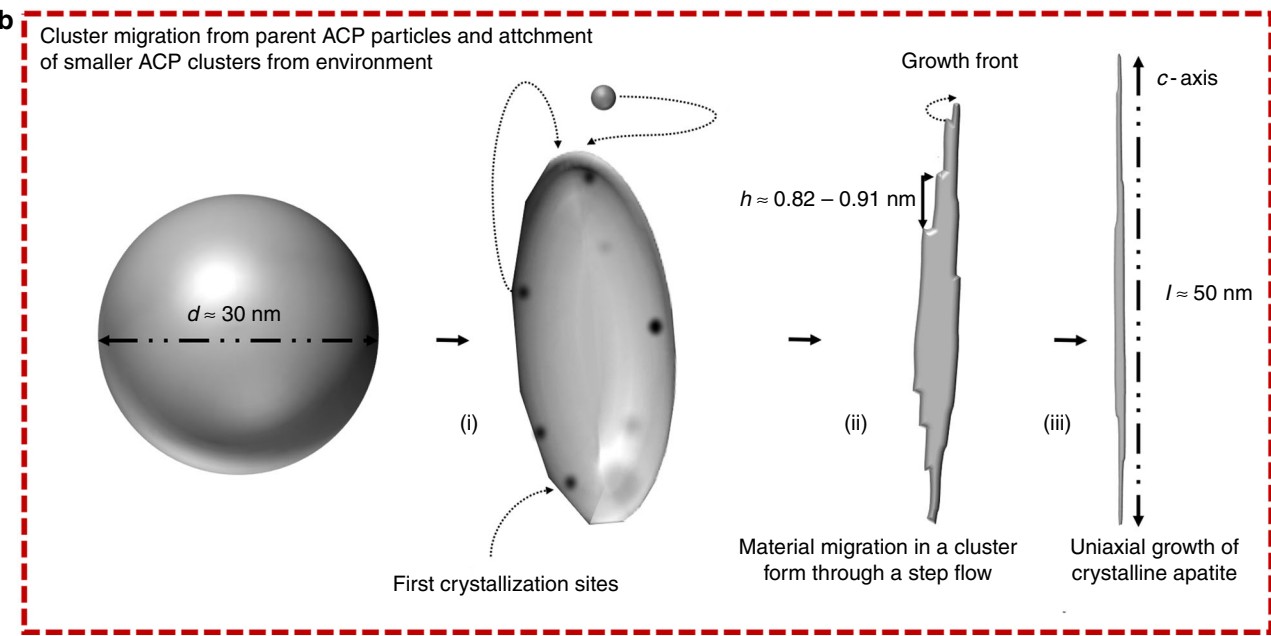

**Fig. 5** Schematic illustration of the proposed ACP to apatite growth mechanism. **a** Formation of ACP particles with spherical morphology, which during aging transform to elongated apatite platelets within the PLLC confinements. **b** Detailed illustration of the growth steps shown in **a**. (i) The ACP particle starts to deform as several crystalline nuclei are formed within the particle. (ii) Steps are formed as a result of a cluster migration in the particle and the crystal starts to adapt a single orientation. Material that is being moved from the sides of the crystal act as seeds for further growth along the c-axis by a layer-by-layer or cluster incorporation mechanism. (iii) Traces of the steps have been left resulting in crystals with non-uniform thickness

Based on the above, we speculate that the growth of apatite within the confinements of collagen fibrils proceeds via a step flow. Despite the similarities of our system to bone, it is evident that the biological environment might also play an important role during bone mineralisation. Collagen and non-collagenous proteins have been shown to be decisive in bone mineralisation by providing confined spaces for precursor penetration and nucleation that aid the growth and orientation of apatite within the gap zones of the collagen matrix[34–36]. Moreover, molecular

dynamics (MD) simulations have shown that the apatite CaP polymorph provides the most energetically favourable interaction with collagen, which was supported by TEM observations[37]. In addition, collagen could also be responsible for the hexagonal 'symmetry breaking' that results in the formation of crystals with platelet morphology[37].

Our proposed crystallisation mechanism provides valuable information for the missing link in the currently accepted stages of bone mineralisation, as outlined in Fig. 6. After the deposition

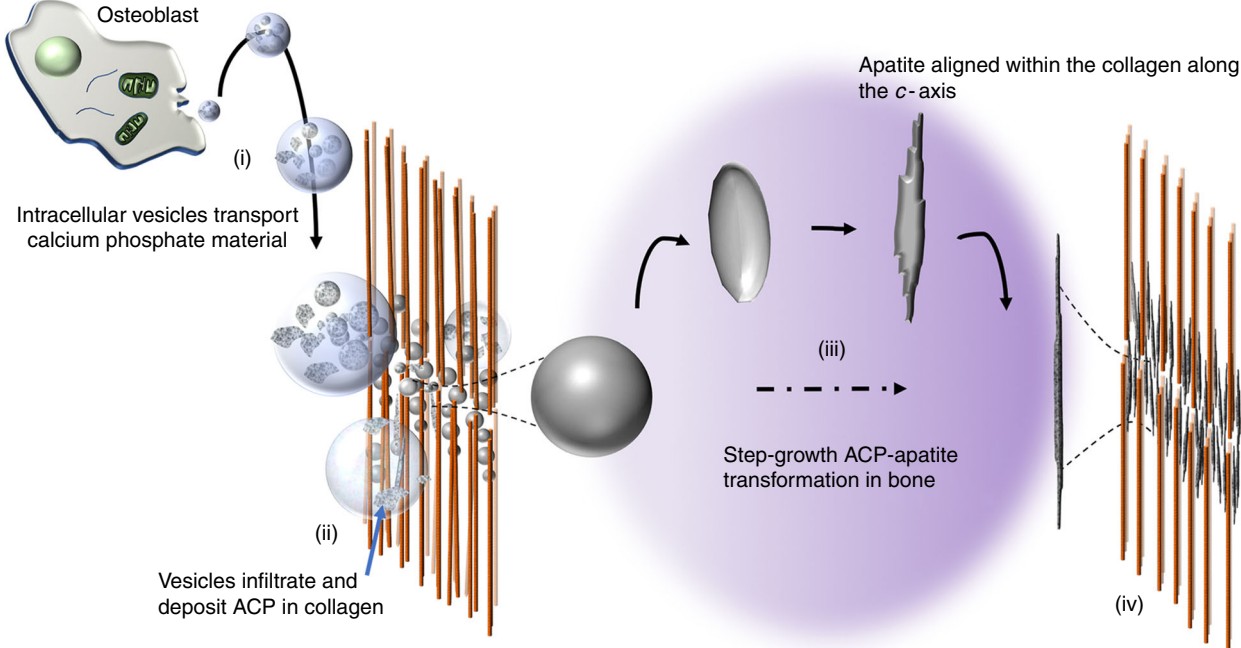

**Fig. 6** Diagram describing a possible bone mineralisation mechanism. Starting from (i), intracellular matrix vesicles bud from osteoblasts and transport ACP precursors as irregular or spherical shaped granules to the gap zones of the collagen matrix (one of the most accepted theories). (ii) The ACP containing vesicles infiltrate the collagen matrix and deposit the ACP granules at the gap zones. (iii) The transformation of the ACP granules into bone apatite along the long-axis of the collagen, via step-flow cluster/ dissolution-growth mechanism. (iv) Fully transformed and matured, mineralised collagen matrix

of ACP granules by vesicles, the ACP phases could transform to apatite according to the growth mechanism presented in this work. A corollary to our hypothesis is the possible explanation it provides for the evolutionary origins of the nanostructure of bone where calcium phosphates seem to be the best choice as the inorganic phase, owing to their thermodynamic property of transforming into apatite nanoplatelets from an easily available and transportable amorphous phase. Decades of bone-research have proven that it is the nanoscale structure and the irregular platelet morphology that drives the intimate contact between collagen and apatite, which directly translates to the strong mechanics and crack-resist behaviour of bone[38,39].

## Discussion

Using a bone-mimetic, apatite-polymer nanocomposite, we have studied the initial crystallisation steps during the transformation of amorphous calcium phosphate to apatite. By employing high-resolution transmission electron microscopy experiments it was revealed that the transformation proceeds via a step growth mechanism where the step dimensions correspond to the sizes of previously theorised pre-nucleation clusters. Considering the irregular shape of apatite crystals, a unified model was proposed for the crystal growth, which proceeds via a cluster migration mechanism through step formation and layer-by-layer. The step growth mechanism, of amorphous calcium phosphate to apatite as described here, could also provide an insight for the origins of the imperfectness in bone apatite (shape and crystallinity), a currently unexplained stage in bone mineralisation.

## Methods
**Synthesis and mineralisation of the bone-mimetic PLLC-CaP composite**. All chemicals were purchased from Sigma-Aldrich. The bone-mimetic nanocomposite was formed by first preparing a hexagonally ordered (H$_1$) lyotropic liquid crystal (LLC) gel. The LLC gel was formed by manually mixing polymerizable triblock co-

polymers (di-acrylate modified Pluronic F127) with 0.84 M Ca$^{2+}$ aqueous calcium phosphate solution (pH = 0.37) and butanol. The gel was mould casted in the form of 2 mm sheets and were photo-crosslinked using UV radiation, which resulted in solid and elastic PLLC[21].

The PLLC matrix was then mineralised with calcium phosphates by increasing the pH within the aqueous domains of the PLLC. This was done by exposing the PLLC sheet to NH$_3$ gas for 3 h at a pressure of 2 bar and at RT inside a sealed autoclave. After the CaP formation, the composites were removed from the NH$_3$ gas chamber and aged at 37 °C, inside an oven. Hence, the PLLC matrix with calcium phosphate ions was in a pressurised autoclave only for the initial ACP particle formation. For aging, the PLLC-ACP composite system was kept in a sealed petri dish in which droplets of Milli-q water were added in order to ensure a humid environment. The total duration of the aging procedure was 3 weeks which is an adequate time period that allowed full crystallisation of the amorphous phase. Four different samples were prepared for the TEM observations; one from the freshly mineralised and unaged PLLC-ACP composite and three aged composite samples, each prepared at 7, 14 and 21 days. For the preparation, the petri dish containing the composite sheet was removed from the oven, and a smaller thin slice was sectioned out using a knife. The thin slice was manually ground in the presence of ethanol to obtain a dispersion. A few drops of the dispersion were dropped onto a C-coated Cu grid and dried. Due to the crosslinked nature of the PLLC, it was difficult to extract or dissolve the polymer phase and separate it from the CaP. Therefore, the TEM specimens contained both PLLC and CaP particles.

**TEM characterisation**. The structural characterisation of the samples was performed using Transmission Electron Microscopy (TEM). High Resolution TEM (HRTEM) experiments were conducted in a Titan 80–300 TEM/STEM equipped with a FEG and a probe Cs corrector (70 pm point resolution in STEM mode), operating at 300 keV and in a Tecnai G2 T20 (0.24 nm point resolution) operating at 200 keV. The TEM specimens also contained the PLLC matrix and some areas having larger fractions of polymer were quite unstable and not easy to image due to beam-induced heat. For this reason, we have not included TEM observations of such areas. However, in some cases (as seen in Fig. 3) remnants of the polymer were present near the apatite crystals outside of the immediate viewing area, which still induced some drift and charging effects of the whole region. It is important to note that the majority of the TEM data presented in this work was obtained from regions with little or no presence of the polymer.

In addition to the potential artefacts from the polymer during imaging, the sensitivity of the inorganic phase should also be discussed. Amorphous CaP is very sensitive under the beam and can decompose rapidly under intense illumination and depends from both the electron dosage, as well as exposure duration. Apatite is also a

beam-sensitive material, especially in the partially crystallised form or as very small crystallites, as shown in Fig. 2 and Fig. 3, where the particles were imaged for > 3 min. Areas of similar morphology were found to completely decompose under continuous illumination for more than 2 min in electron dosages of > $10^6$ enm$^{-2}$s$^{-1}$. The fully aged apatite crystals (Fig. 4) were not observed to be beam sensitive and no apparent damage or structural differences were observed while imaging independent of the electron dosage. In the cases of ACP, small crystals and partially crystallised particles, common techniques to minimise the beam damage were employed. The images were focused in adjacent areas away from the area of interest in which the electron dose was also lowered. For recording the images, short exposure times were employed (0.5–0.8 s) and in the cases where more than one image was recorded, the area of interest was moved out of the viewing axis. In most of the cases, 1 or 2 images of the amorphous particles per area in the low magnification range (less than ×100k) was acquired. In the cases as in Figs. 2 and 3 where the material transformation was monitored, the images were recorded two minutes apart and between recordings the area was moved out of the viewing axis. The electron dosage varied between 15,600 (in low magnifications of ~×100k) and $8.86 \times 10^5$ enm$^{-2}$s$^{-1}$ in higher magnifications (~×600k).

**XRD measurements**. XRD was used to follow and analyse the transformation process of the ACP nanoparticles into apatite crystals within the PLLC matrix. The composite samples for XRD were prepared in the form of thin rectangular sheets with size of $1 \times 20 \times 20$ mm$^3$ and the mineralisation was performed in the NH$_3$ chamber as mentioned above. The sheets were analysed immediately post mineralisation, as well as at different time points during the aging process. The aging was performed similar to the samples analysed using TEM, i.e. at 37 °C for a maximum time period of 1 week. A main difference with XRD was that the sample was analysed as such, i.e. the whole sheet was analysed without grounding or post processing. This resulted in a XRD signal from polyethylene oxide groups present in the main chain of the Pluronic F127 polymer. The instrument was a Siemens D5000 ($\lambda = 1.54$ Å, exposure time 1 s/step over 5800 steps between $2\theta = 3–60°$). The acquired XRD data was filtered using the Savitzky–Golay method in MATLAB 2016b.

## Data availability

All data are available in the article and from the authors upon reasonable request.

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

## Acknowledgements

A.L. acknowledges Chalmers Area of Advance 'Nanoscience and Nanotechnology' for the financial support. A.K.R. and M.A would like to thank the Wallenberg Foundation and the Chalmers Area of Advance 'Materials Science'.

## Author contributions

A.L. performed the TEM experiments and the GPA analysis. A.K.R. produced the samples and performed the XRD measurements. A.L. and A.K.R. analysed the data. M.H. and M.A. supervised the project. All authors wrote the manuscript.

## Additional information

**Competing interests:** The authors declare no competing interests.

