## [Peer Review File · Nature Communications]

Reviewers' Comments:

Reviewer #1:

Remarks to the Author:

This manuscript presents the early stages of the transformation from amorphous calcium phosphate into crystalline hydroxyapatite studied by HRTEM. It defends that the crystallization process starts by ionic dissociation/migration rather than dissolution/precipitation. Then it proposes a mix mechanism for the next crystallization stage (step flow plus layer by layer) and theorize that the environment does not determine the crystallization process.

The originality of the paper relies on the study of the first amorphous to crystalline transformation that occurs in the calcium phosphate system. It defends that the transformation is due to ionic dissociation/migration rather than dissolution/precipitation. But this statement is based on the morphology on the surface of the particle only. No measurements of the ion concentration or pH are provided. It challenges former published papers (ie. NATURE COMMUNICATIONS | DOI: 10.1038/ncomms2490) where this transformation relies in the consumption of ions from the media. And although it seems as the most compatible explanation with the microscopy observation, it should remain only as a proposal since it is not contrasted. However, for further crystallization steps a complementary precipitation process is accepted and multiple crystal growth mechanisms are discussed and accepted.

Line 182, " ...It is important to note that no intermediate CaP phases have been observed during the crystallization. This, however, cannot confirm the absence of any metastable phase prior to the formation of initial crystalline domains at the edges of the ACP particle...." Octacalcium phosphate has been frequently described as an intermediate phase due to close structural relationship to hydroxyapatite and because it nucleates and crystallizes more easily than hydroxyapatite. A specific analysis and explanation would be required to dismiss the presence of this phase in any of the presented studies

Line 323. ... ACP to apatite transformation is an intrinsic property of the material; which, stripped of any biological influence, can be treated as an inorganic crystal.... This is not a "happy" sentence since later and earlier on the paper the role of the biological component is admitted. And it should be rewritten considering the stability of calcium phosphate phases dictated by thermodynamics.

- Data & methodology:

Lines 100-117, describes the method partially and refers to the methods section for details. These details are then not provided. Times of sample extraction during the three weeks are missing and how this is done since crystallization is developed in a pressured system. The main texts mention 2 minutes as an example

Line 448 CaP. "... Therefore, the TEM specimens contained both the PLLC and the CaP particles. However, care was taken to avoid artefacts emerging from the PLLC matrix...." Please explain the procedures to "take care": Amorphous CaP is very sensitive to ion beam an Moire fringes as in supplementary fig 2 are expected. Please describe whether measurements to avoid induced deformation have been taken.

- Conclusions:

Line 419 "...we hypothesized that the crystal growth mechanism from ACP to apatite is inherent to calcium phosphates and appears to be independent of the environment." As in the main text, this sentence should mention thermodynamics rather than "intrinsic"

Line 421 "These results also provide a possible answer for the origins of the structural imperfectness in bone-hydroxyapatite, a currently unexplained stage in bone-mineralization..." That is a discussion point but not a conclusion extracted from the results

- Suggested improvements:

PH and Ion concentration measurements that validate the proposed ionic dissociation/migration mechanisms rather than dissolution/precipitation would be required.

The HRTEM study of the possible presence of octacalcium phosphate should be provided at least in the supplementary material

References are appropriate and the abstracts describes the content of the manuscript.

Luis M. Rodríguez-Lorenzo

Reviewer #2:

Remarks to the Author:

It has already been proposed that the conversion of amorphous calcium phosphate (ACP) to hydroxyapatite (HA) can play a very important role in bone mineralization. This process has already been studied (see, for example refs [1-3] and others in the paper). The main novel point of this paper is the use of high-resolution transmission electron microscopy to follow this transformation in a matrix that tries to mimic collagen. Perhaps the authors would like to comment more on the possible effect of the interactions between the material and the electron beam on the observations. As the time evolution of the samples is in some cases followed in the microscope, will interaction with the beam affect the observations? The authors seems to suggest that it favors evaporation-condensation (line 275) that will act as solution-precipitation in liquid media. If this is the case then could also provide enough energy (atomic mobility) to favor crystallization? Can it form the disordered surface layers observed (or could this be an artifact)? In the liquid media the crystal is in equilibrium with a solution of a given composition, in the microscope the crystal is under dynamic vacuum, so to speak, ions are being removed from the surrounding gas. Are the two systems comparable? How stable is HA under the beam? HA will decompose when heated, is this a problem in the microscope?

In this system there is no chemical interaction between the matrix and the calcium phosphate. The authors propose that it is a good analogue to the biological system and therefore suggest that the conversion of ACP to HA is an "intrinsic" property of the material perhaps affected by the close confinement in the matrix. It is true that conversion of ACP to HA has been observed in solution and that there are some parallels between the morphology of the HA crystals in mineralized bone and those observed here. However, there is evidence that that collagen plays an active role in the nucleation of HA and that non collagenous components regulate HA nucleation and collagen mineralization (see for example [4-8] between many others. It has also been suggested that octacalcium phosphate can act as a precursor of HA[9, 10]. Furthermore nucleation and growth of HA is a complex phenomena that will depend on the pH and composition of the media. It has already been observed that the growth habit of hydroxyapatite in solution or some synthetic matrices is in the form of needles or platelets. Perhaps the authors could comment more on the existing literature on the anisotropy of interfacial energy and growth rates in HA.

Overall, I think that this work has some high quality data that deserves publication. However, in my opinion the analysis of the relevance to bone mineralization should undergo a major revision where the findings of this paper are put in context with all previous literature on bone mineralization and the role of collagen and other non-collagenous components.

1. Boskey, A.L. and A.S. Posner, Conversion of Amorphous Calcium Phosphate to Microcrystalline Hydroxyapatite - Ph-Dependent, Solution-Mediated, Solid-Solid Conversion. *Journal of Physical Chemistry*, 1973. 77(19): p. 2313-2317.
2. Harries, J.E., D.W.L. Hukins, C. Holt, and S.S. Hasnain, Conversion of Amorphous Calcium-Phosphate into Hydroxyapatite Investigated by Exafs Spectroscopy. *Journal of Crystal Growth*, 1987. 84(4): p. 563-570.
3. Tao, J.H., H.H. Pan, Y.W. Zeng, X.R. Xu, and R.K. Tang, Roles of amorphous calcium phosphate and biological additives in the assembly of hydroxyapatite nanoparticles. *Journal of Physical Chemistry B*, 2007. 111(47): p. 13410-13418.
4. Baht, G.S., G.K. Hunter, and H.A. Goldberg, Bone sialoprotein-collagen interaction promotes

hydroxyapatite nucleation. *Matrix Biology*, 2008. 27(7): p. 600-608.

5. Boskey, A.L., Mineral-Matrix Interactions in Bone and Cartilage. *Clinical Orthopaedics and Related Research*, 1992(281): p. 244-274.

6. Nudelman, F., K. Pieterse, A. George, P.H.H. Bomans, H. Friedrich, L.J. Brylka, P.A.J. Hilbers, G. de With, and N.A.J.M. Sommerdijk, The role of collagen in bone apatite formation in the presence of hydroxyapatite nucleation inhibitors. *Nature Materials*, 2010. 9(12): p. 1004-1009.

7. Rhee, S.H., J.D. Lee, and J. Tanaka, Nucleation of hydroxyapatite crystal through chemical interaction with collagen. *Journal of the American Ceramic Society*, 2000. 83(11): p. 2890-2892.

8. Chen, Y., B.S. Bal, and J.P. Gorski, Calcium and Collagen Binding-Properties of Osteopontin, Bone Sialoprotein, and Bone Acidic Glycoprotein-75 from Bone. *Journal of Biological Chemistry*, 1992. 267(34): p. 24871-24878.

9. Brown, W.E., Crystal growth of bone mineral. *Clinical orthopaedics and related research*, 1966. 44: p. 205-220.

10. Bodier-Houlle, P., P. Steuer, J.C. Voegel, and F.J.G. Cuisinier, First experimental evidence for human dentine crystal formation involving conversion of octacalcium phosphate to hydroxyapatite. *Acta Crystallographica Section D-Biological Crystallography*, 1998. 54: p. 1377-1381.

Reviewer #3:

Remarks to the Author:

The introduction suggests that we will get a real insight into what happens mechanistically in vivo, but then it turns out another in vitro study. Given the fact that in this study a synthetic scaffold is used, the importance of the work is a level lower than the studies using collagen as the template (Gower, Beniash, Sommerdijk, Nassif, Tay).

In addition, the TEM experiments are conventional (dry)TEM experiments – in contrast to the work of Sommerdijk, Nassif and Tay), so they provide very limited information even on the in vitro mineralization process.

Essentially, the authors show the solid state transformation of ACP to apatite in a synthetic environment, in fact in vacuum (which again may affect the transformation).

I therefore must conclude that the paper is not of sufficient interest for the broad audience of *Nature Communications*, and should better be submitted to a more specialized journal. Moreover, the authors make several poorly informed statements, which are in fact dangerous to the field if they are published like this. So I cannot support publication in this form.

Examples:

Line 34 – there is no hard evidence that ACP is delivered by intracellular vesicles

Line 40 – not all these papers refer to biological systems. Although all these papers are relevant, some are biomimetic systems

Line 41 - "while there is a continuous confirmation ..." I am not sure what is meant by this. Can the authors give the examples? I am not sure there is a continuous stream of evidence for this.

Figure 6. It is a bit bold to state that this is "the current model for bone mineralization". The community will agree there are many things unclear.

And more technically:

The authors suggest that "taking the images 2 min apart" may limit beam damage. This is not true, beam damage is mainly depending on the total electron dose.

Response to Referees' comments:

Reviewer #1

Point 1:

“This manuscript presents the early stages of the transformation from amorphous calcium phosphate into crystalline hydroxyapatite studied by HRTEM. It defends that the crystallization process starts by ionic dissociation/migration rather than dissolution/precipitation. Then it proposes a mix mechanism for the next crystallization stage (step flow plus layer by layer) and theorize that the environment does not determine the crystallization process. The originality of the paper relies on the study of the first amorphous to crystalline transformation that occurs in the calcium phosphate system.”

Authors comment:

To this introductory paragraph we do not have any comments other than we do not defend dissociation/migration over dissolution/precipitation but rather propose the observed mechanism (dissociation/migration) as a possible crystallization pathway from the initial amorphous calcium phosphate (ACP) to bone-like apatite. This is explained in more detail in Point 2 below.

Point 2:

“It defends that the transformation is due to ionic dissociation/migration rather than dissolution/precipitation. But this statement is based on the morphology on the surface of the particle only. No measurements of the ion concentration or pH are provided. It challenges former published papers (i.e. NATURE COMMUNICATIONS | DOI: 10.1038/ncomms2490) where this transformation relies in the consumption of ions from the media. And although it seems as the most compatible explanation with the microscopy observation, it should remain only as a proposal since it is not contrasted. However, for further crystallization steps a complementary precipitation process is accepted and multiple crystal growth mechanisms are discussed and accepted.”

Authors comment:

This is an important issue raised by the reviewer. The ionic cluster migration of material during the transformation of ACP to apatite is the only mechanism that we observed during our TEM experiments. Given that the experiment is *ex situ*, the dissolution/(re)precipitation mechanism is not possible to observe, since the presence of an aqueous environment is crucial for this mechanism to occur. Any material loss inside the

microscope is probably induced by the electron beam, which cannot be related to a dissolution process. Our primary focus with this study was to elucidate the morphological and structural changes taking place in the ACP particle and its eventual transformation to apatite. We do not attempt to contradict former published work that incorporates the dissolution-precipitation mechanism for the ACP-apatite transformation. On the contrary, we acknowledge and refer to those results (References 13-20). Furthermore, we assert that the actual ACP-apatite transformation might proceed via a combination of pathways that might include both the dissolution-precipitation and the proposed cluster migration mechanisms. This has now been clearly stated in the discussion of the revised manuscript in Lines 264-275 by stating that *“it cannot be the sole mechanism...strongly indicates that a dissolution mechanism also takes place under the presence of water”* and *“...it is generally accepted that a dissolution - re-precipitation mechanism takes place in apatite transformation....but also has been shown in situ that such a mechanism occurs simultaneously with stepped spiral growth”*.

During the mineralization of the materials described in our work, the calcium and phosphate ions are precipitated from a supersaturated aqueous media, which is located between the confined nanodomains of the PLLC matrix. The precipitation is enabled by a rapid increase in pH within the nanodomains using ammonia gas. The mineralization process is necessary for the initial formation of the ACP particles that form within the confined domains between micellar nanofibrils. However, after this step, the PLLC-ACP composite is removed from the ammonia gas chamber and placed in a humidified chamber at 37° C for the aging process. Therefore, the initial ionic concentration is available only for the ACP formation following which there is no additional supply of calcium and phosphate ions during the aging process. This demonstrates that the crystallization or ACP transformation does not occur in the presence of a continuous supply of ions.

Relevant text has been added in the Methods section of the revised manuscript to include all details of the experiments (*see also Point 5*).

Additional changes in the manuscript to clarify further that we do not impose the dissociation/migration mechanism as the sole mechanism taking place, have been added. Specifically, in:

- i) Line 18: replaced “intrinsic” for *“possible crystallization mechanism and could be characteristic of calcium phosphates from a thermodynamic perspective and might be unrelated to the environment”*
- ii) Line 273: “*ex situ*” term was added

- iii) Line 274: added text “...a dissolution – re-precipitation mechanism could be a parallel transformation mechanism in an aqueous environment.” The earlier phrase “equivalent process.... electron beam radiation” was removed since it was not a suitable expression. Material lost from beam damage cannot be re-attached to the growing material (see also Point 2 of Reviewer #2)
- iv) Line 301, added text: “The structural and morphological transformation of ACP to apatite is a complex procedure, and the material migration through a step flow that was observed here could be a part of the complex transformation process that is present in sequence or simultaneously with additional mechanisms (such as dissolution and precipitation).”

Point 3:

“Line 182, ...It is important to note that no intermediate CaP phases have been observed during the crystallization. This, however, cannot confirm the absence of any metastable phase prior to the formation of initial crystalline domains at the edges of the ACP particle....” Octacalcium phosphate has been frequently described as an intermediate phase due to close structural relationship to hydroxyapatite and because it nucleates and crystallizes more easily than hydroxyapatite. A specific analysis and explanation would be required to dismiss the presence of this phase in any of the presented studies”

Authors comment:

Octacalcium phosphate (OCP) has been proposed as an intermediate phase through which ACP transforms to apatite, especially in the case of human dentine crystals as previously demonstrated by [(1) P. Bodier-Houlle *et al.* First Experimental Evidence for Human Dentine Crystal Formation Involving Conversion of OCP to HAp. *Acta Cryst.* **D54**, 1377 (1998) (2) J. Reyes-Gasga *et al.*, Aberration Corrected TEM Study of the CDL Defect in Human Tooth Enamel Crystals. *Microsc. Microanal.* **22**, 1047 (2016) (3) Crane, N.J. *et al.* Raman spectroscopic evidence for octacalcium phosphate and other transient mineral species deposited during intramembranous mineralization. *Bone* **39**, 434-442 (2006)] and in specific biomimetic nucleation of CaP [Habraken W.J.E.M. *et al.* Ion-association complexes unite classical and non-classical theories for the biomimetic nucleation of calcium phosphate. *Nat. Comm.* **4**, 1507 (2013)]. On the other hand, there are references stating that OCP was not present as a transient phase or if present was not detectable in *in-vivo* studies by Mahamid J. *et al.*, Amorphous calcium phosphate is a major component of the forming fin bones of zebrafish: Indications for an amorphous precursor phase. *Proc. Natl Acad. Sci. USA* **105**, 12748–12753 (2008). The subject has been controversial in the field and this can be clearly depicted by the two *Bone* editorials by

M.D. Grynpas [Grynpas M.D. & Omelon S., Transient precursor strategy or very small biological apatite crystals? *Bone* **41**, 162 (2007)] and S. Weiner [Weiner S. Transient precursor strategy in mineral formation of bone. *Bone* **39**, 431 (2006)] and the references within show that “OCP-like” or “modified OCP” structures were identified.

During the evaluation of our TEM data, a significant amount of time was focused on trying to identify the OCP phase during the initial transformation stages. In our system the spherical ACP particles begin to exhibit crystalline nuclei (such as those in Fig. 1d) and small crystals (Fig. 3) with d-spacings and atomic arrangements that correspond to apatite instead of the triclinic OCP system. In the case of the partially crystallised particles (such as in Fig. 2) the apatite-specific (1-100) atomic planes are resolved even on the initial crystalline area, indicating the absence of OCP even during the first stages of ACP transformation (which here transform directly to apatite). If indeed OCP was present before the formation of initial crystallisation nuclei (meaning that the nuclei are OCP that transforms to apatite), it would be extremely difficult to observe the same unless the crystallites were oriented along an axis where their (100) planes are resolved (the only planes with significant difference from apatite).

Since TEM could not establish the presence of OCP, we performed additional XRD experiments on the PLLC-ACP composite during the initial ACP formation and followed the measurements during the subsequent aging process for 5 days. In the XRD diffractograms, the main peaks for OCP i.e. at low angles were absent, indicating that no OCP was present during the transformation or possibly that the amount was so small that it was not detectible.

In order to accommodate the additional information, the text in Lines 182-185 *“It should be emphasized that no intermediate CaP phases have been observed during the crystallisation. This; however, cannot confirm the absence of any metastable phase prior to the formation of initial crystalline domains at the edges of the ACP particle. Although octacalcium phosphate (OCP) has been widely proposed to act as a transient metastable phase in mineralisation, its presence is very difficult to identify with TEM due to its high similarity to apatite. The two CaP phases differ only in the interplanar spacing of the (100) OCP / (10 $\bar{1}$ 0) apatite planes. In our observations, the material is being transformed from amorphous to partially-crystallised to fully crystallised and even in the initial crystalline areas, the unique to apatite (10 $\bar{1}$ 0) planes are being resolved, indicating the absence of OCP. To further investigate the possible OCP formation before apatite, we performed XRD measurements on freshly made composites (unaged) and monitored them up to 5 days of aging in humid environment at 37 °C. As can be seen in the XRD diffractograms in Extended Data Figure 3, the strongest peaks*

of OCP, i.e. (100) and (010) at 2 Theta 4.747° and 9.744° respectively, are absent thus confirming the TEM observations that ACP transforms directly to apatite.” has been moved to Line 193 (highlighted text). XRD results are also added as Extended Data Figure 3. Relevant experimental information regarding the XRD experiments has been added in the **Methods** Section.

Point 4:

“Line 323. ... ACP to apatite transformation is an intrinsic property of the material; which, stripped of any biological influence, can be treated as an inorganic crystal.... This is not a “happy” sentence since later and earlier on the paper the role of the biological component is admitted. And it should be rewritten considering the stability of calcium phosphate phases dictated by thermodynamics.”

Authors comment:

We thank the reviewer for this comment and have re-written the text in Line 301 “The structural and morphological transformation of ACP to apatite is a complex process. The step flow migration of material, as observed here, could be part of the transformation process in combination with additional mechanisms such as dissolution and precipitation. In the presence of water, a dissolution – re-precipitation mechanism possibly occurs in coordination with the above steps. The crystal growth mechanism from ACP to apatite follows the thermodynamically driven phase transformation that is governed by the solubility of the CaP phases. For this transformation, a humid environment and ambient conditions are necessary and the orientation and location of the crystals are hindered by the polymeric network’s space confinement.” and Line 322 “The transformation of ACP to apatite is a process that is common in both synthetic and biological systems. This is expected since apatite is the most thermodynamically stable CaP phase under physiological conditions. In our specific case, the resulting apatite crystals are similar to the apatite found in bone, both in terms of morphology and crystallinity” and an appropriate reference has been added (Reference 32, Wang L. & Nancollas G.H. Calcium Orthophosphates: Crystallization and dissolution. *Chem. Rev.* **108(11)**, 4628 (2008).). Additional text relevant to the importance of the biological environment has also been added in the Section **Relevance to Bone Mineralization**.

Point 5:

“Data & methodology: Lines 100-117, describes the method partially and refers to the methods section for details. These details are then not provided. Times of sample extraction during the three weeks are missing and how this is done since crystallization is developed in a pressured system. The main texts mention 2 minutes as an example.”

Authors comment:

We thank the reviewer for pointing out lack of clarity in the described methods. The text in Lines 433-449 (Methods) has been rewritten in order to clarify the various procedures. We would also like to note that the 2 minutes period refers to the time between imaging of the particles and does not refer to the specimen preparation procedure. The following text has been added in the Methods section to clarify the sample preparation for the TEM analysis:

“After the CaP formation, the composites were removed from the NH₃ gas chamber and aged at 37 °C, inside an oven. Hence, the PLLC matrix with calcium phosphate ions was in a pressurised autoclave only for the initial ACP particle formation. For aging, the PLLC-ACP composite system was kept in a sealed petri dish in which droplets of Milli-q water were added in order to ensure a humid environment. The total duration of the aging procedure was 3 weeks which is an adequate time period that allowed full crystallization of the amorphous phase. Four different samples were prepared for the TEM observations; one from the freshly mineralized and unaged PLLC-ACP composite and three aged composite samples, each prepared at 7, 14 and 21 days. For the preparation, the petri dish containing the composite sheet was removed from the oven, and a smaller thin slice was sectioned out using a knife. The thin slice was manually ground in the presence of ethanol to obtain a dispersion. A few drops of the dispersion were dropped onto a C coated Cu grid and dried. Due to the cross-linked nature of the PLLC, it was difficult to extract or dissolve the polymer phase and separate it from the CaP. Therefore, the TEM specimens contained both PLLC and CaP particles.”

Point 6:

“Line 448 CaP.”... Therefore, the TEM specimens contained both the PLLC and the CaP particles. However, care was taken to avoid artefacts emerging from the PLLC matrix....” Please explain the procedures to “take care”: Amorphous CaP is very sensitive to ion beam and Moiré fringes as in supplementary fig 2 are expected. Please describe whether measurements to avoid induced deformation have been taken.”

Authors comment:

The text in Line 448 has been changed and following text relevant to the sensitivity of the composite (both for the polymer and the inorganic component) has been added at the end of the **Methods** Section in the manuscript (Line 444): *“The TEM specimens also contained the PLLC matrix and some areas having larger fractions of polymer were quite unstable and not easy to image due to beam-induced heat. For this reason, we have not included TEM observations of such areas. However, in some cases (as seen in Fig. 3) rem-*

nants of the polymer was present near the apatite crystals outside of the immediate viewing area, which still induced some drift and charging effects of the whole region. It is important to note that the majority of the TEM data presented in this work was obtained from regions with little or no presence of the polymer. In addition to the potential artefacts from the polymer during imaging, the sensitivity of the inorganic phase could also be discussed. Amorphous CaP is very sensitive under the beam and can decompose rapidly under intense illumination. Apatite is also a beam sensitive material, especially in the partially crystallised form or as very small crystallites, as shown in Fig. 2 and Fig. 3, where the particles were imaged for >3 mins. The fully aged apatite crystals (Fig. 4) was not observed to be beam-sensitive and no apparent damage or structural differences were observed while imaging. In the cases of ACP, small crystals and partially crystallised particles, common techniques to minimize the beam damage were employed. The images were focused in adjacent areas away from the area of interest in which the electron dose was also lowered. For recording the images, short exposure times were employed (0.5-0.8 s) and in the cases where more than one image was recorded, the area of interest was moved out of the viewing axis. In most of the cases, 1 or 2 images of the amorphous particles per area in the low magnification range (less than x100k) was acquired. In the cases as in Fig. 2 and 3 where the material transformation was monitored, the images were recorded two minutes apart and between recordings the area was moved out of the viewing axis.”

Point 7:

“Conclusions: Line 419 “...we hypothesized that the crystal growth mechanism from ACP to apatite is inherent to calcium phosphates and appears to be independent of the environment.” As in the main text, this sentence should mention thermodynamics rather than “intrinsic””

Authors comment:

The text in Line 419 has been removed and relevant changes have been added in the main text as following (see also Point 4). “*thermodynamic property of transforming into apatite nanoplatelets from an easily available and transportable amorphous phase.*”

Point 8:

“Line 421 “These results also provide a possible answer for the origins of the structural imperfection in bone-hydroxyapatite, a currently unexplained stage in bone-mineralization...” That is a discussion point but not a conclusion extracted from the results”

Authors comment:

We thank the reviewer for emphasizing this point. The text in Line 421 has been rephrased as “*The step-growth mechanism, of amorphous calcium phosphate to apatite as described here, could*

also provide an insight for the origins of the structural imperfectness in bone-apatite (shape and crystallinity), a currently unexplained stage in bone-mineralization.”

Point 9:

“Suggested improvements: PH and Ion concentration measurements that validate the proposed ionic dissociation/migration mechanisms rather than dissolution/precipitation would be required. The HRTEM study of the possible presence of octacalcium phosphate should be provided at least in the supplementary material. References are appropriate and the abstracts describes the content of the manuscript.”

Authors comment:

We thank the reviewer for the suggestions for additional experiments. To analyse the variations in pH during aging, one should necessarily probe within the nanodomains of the polymeric network (where ACP resides). Given that the spacing of the polymer fibrils is 10 nm, quantitative analysis of the changes in ion concentrations within the nanoscale domains can be very difficult to estimate. It is certainly possible to measure the pH while the ACP particles are freely dispersed in an aqueous solution without the polymer, which however compromises the biologically relevant factor of geometric confinements. Measurements on the ionic concentration also presented an additional obstacle, that of continuous and simultaneous dissolution/ re-precipitation, something which would not give an accurate presentation of the variations in concentration.

Regarding the presence of OCP in our system, the use of HRTEM was not adequate to confirm the presence of OCP. Instead, additional XRD experiments were performed to follow the ACP-apatite transformation, as explained in our response to Point 3.

Reviewer #2

“It has already been proposed that the conversion of amorphous calcium phosphate (ACP) to hydroxyapatite (HA) can play a very important role in bone mineralization. This process has already been studied (see, for example refs [1-3] and others in the paper). The main novel point of this paper is the use of high-resolution transmission electron microscopy to follow this transformation in a matrix that tries to mimic collagen.”

Point 1:

“Perhaps the authors would like to comment more on the possible effect of the interactions between the material and the electron beam on the observations. As the time evolution of the

samples is in some cases followed in the microscope, will interaction with the beam affect the observations?”

Authors comment:

We thank the reviewer for this important comment. We have added relevant text in the Methods Section, addressing the electron beam effects in the various CaP morphologies and phases we observe in our study. The text in Line 448 has been rephrased and text relevant to the sensitivity of the composite (both for the polymer and the inorganic component) has been added at the end of the **Methods** Section in the manuscript (Line 444): *“The TEM specimens also contained the PLLC matrix and some areas having larger fractions of polymer were quite unstable and not easy to image due to beam-induced heat.... the majority of TEM data... was obtained from regions with little or no presence of the polymer. In addition to the potential artefacts from the polymer during imaging, the sensitivity of the inorganic phase could also be discussed ... the area was moved out of the viewing axis.”*

Point 2:

“The authors seems to suggest that it favors evaporation-condensation (line 275) that will act as solution-precipitation in liquid media. If this is the case then could also provide enough energy (atomic mobility) to favor crystallization? Can it form the disordered surface layers observed (or could this be an artifact)? In the liquid media the crystal is in equilibrium with a solution of a given composition, in the microscope the crystal is under dynamic vacuum, so to speak, ions are being removed from the surrounding gas. Are the two systems comparable? How stable is HA under the beam? HA will decompose when heated, is this a problem in the microscope?”

Authors comment:

As mentioned in an earlier comment (Point 2, Reviewer 1), the phrase in Line 275 *“equivalent process.....electron beam radiation”* was removed since it was not a suitable expression for explaining the process. The dissolution/precipitation mechanism that might take place while ACP transforms to apatite cannot be monitored in *ex situ* experiments. If indeed material is lost due to beam damage, then the lost material cannot be re-accumulated in the crystal structure under vacuum. The term “equivalent process” referred to the fact that the material might be lost in the form of clusters rather than ions.

While it is possible that the electron beam provides the energy for atomic mobility that might promote the extension of a pre-existing crystalline region, in our experiments we did not observe the generation of any new crystalline material (or domains) within the

amorphous matrix of the ACP particles as induced by the electron beam. In all our observations the crystalline regions were present and easily detectable even in low magnifications (>x 80k) due to the contrast difference with the amorphous material. Moreover, taking into consideration the complex atomic structure of the apatite crystal, it seems unlikely that crystallisation was induced by the beam. On the other hand, deformation and decomposition (beam damage) can take place under continuous illumination for prolonged time periods (>3-4 minutes).

We acknowledge that the two systems, liquid media and dynamic vacuum, cannot be directly compared. However, the structural evolution from amorphous spherical particles to bone-like platelets correlate well to the results from previously published works (References [1-3] provided in the reviewer's comments that are added also here at the end). Moreover, the step-flow growth mechanism has also been previously observed using *in-situ* AFM with lower resolution than TEM (*in situ* AFM – References [15-19] in our manuscript). Due to these similarities we can conclude that despite the differences, the results from our work can be extended as a general pathway for the ACP to apatite transformation mechanism.

We have added new (highlighted) text in Methods Section (response in Point 1) of our revised manuscript regarding the sensitivity of the aged apatite crystals to the electron beam. The disordered surface layer that was observed in crystals and particles at different aging times is clearly evident in the fully aged crystals, which are very stable under the electron beam. Note that these disordered surfaces have also been observed in calcified biominerals (Reference [30] in the manuscript). For this reason, we do not believe that this layer is an artefact.

Point 3:

“In this system there is no chemical interaction between the matrix and the calcium phosphate. The authors propose that it is a good analogue to the biological system and therefore suggest that the conversion of ACP to HA is an “intrinsic” property of the material perhaps affected by the close confinement in the matrix. It is true that conversion of ACP to HA has been observed in solution and that there are some parallels between the morphology of the HA crystals in mineralized bone and those observed here. However, there is evidence that that collagen plays an active role in the nucleation of HA and that non collagenous components regulate HA nucleation and collagen mineralization (see for example [4-8] between many others.”

Authors comment:

The reviewer has raised an important point here that relates to the earlier point made by reviewer 1. The structural evolution has now been explained in term of thermodynamics rather than using the term “intrinsic” when referring to the transformation of ACP to apatite. The relevant text in the main text has been changed to “*The transformation of ACP to apatite is a process that is common in both synthetic and biological systems. This is expected since apatite is the most thermodynamically stable CaP phase under physiological conditions. In our specific case, the resulting apatite crystals are similar to the apatite found in bone, both in terms of morphology and crystallinity*”. We refer to Point 2 of the first Reviewer.

Regarding the role of the biological component, we kindly refer to our response to Point 4 of the Reviewer 1 for changes in the main text in Lines 322. Additional text has been also added in the **Relevance to bone mineralization** Section, “*Despite the similarities of our system to bone, it is evident that the biological environment might also play an important role during bone-mineralization. Collagen and non-collagenous proteins have been shown to be decisive in bone mineralization by providing confined spaces for precursor penetration and nucleation that aid the growth and orientation of apatite. Moreover, molecular dynamics (MD) simulations have shown that the apatite CaP polymorph provides the most energetically favourable interaction with collagen, which was supported by TEM observations. In addition, collagen could also be responsible for the hexagonal “symmetry breaking” that results in the formation of crystals with platelet morphology*” Additional reference, number 37, has been cited here, Tao J. et al., *Energetic basis for the molecular-scale organization of bone*. PNAS 112(2), 326 (2015).

Point 4:

“It has also been suggested that octacalcium phosphate can act as a precursor of HA[9, 10].”

Authors comment:

This issue has been also raised by Reviewer 1. We refer to our earlier response Point 3 from Reviewer 1.

Point 5:

“Furthermore nucleation and growth of HA is a complex phenomenon that will depend on the pH and composition of the media. It has already been observed that the growth habit of hydroxyapatite in solution or some synthetic matrices is in the form of needles or platelets. Perhaps the authors could comment more on the existing literature on the anisotropy of interfacial energy and growth rates in HA.”

Authors comment:

We agree with the reviewer that the pH, composition of the media and the degree of supersaturation are crucial aspects for the ACP nucleation and its transformation to apatite. The synthesis conditions that were used in this work were derived from our previous works that resulted in the formation of oriented bone-like apatite platelets. In the text we refer to these works and their experimental Sections (Line 91) and the already published results (Line 91-97) (References 21-24). In general, the growth of apatite from ACPs have been shown to attain a single orientation with platelet geometry both in aqueous solution and in confined domains. The platelet morphology is always attained in biological systems, but the reasons as to why this happens are still unknown (Dorozhkin S.V. Epple M. Biological and Medical Signification of CaP. *Angew. Chem. Int. Ed.* **41**, 3130 (2002) and the references therein). This morphology cannot be attributed to the crystallographic symmetry of a hexagonal unit cell but might be related to the spatial restriction during bone formation (Kobayashi T. et al., Morphological variation of HAp grown in aqueous solution based on SBF. *Cryst. Eng. Comm.* **14**, 1143 (2012)). While the symmetry of the hexagonal structure is violated upon the platelet formation, the crystallographic faces present in the bulk hexagonal prism morphology of apatite (that can be obtained in equilibrium conditions) are present in the platelets as well. The (0001) surface exhibits the lowest surface energy value and highest growth rate but is not the most exposed (on the top view) and the platelets are bounded from the sides with {01-1} faces that is produced by the hexagonal symmetry (Chiatti F., Delle Piane M. Ugliengo P., Corno M. Water at HAp surfaces: the effect of coverage and surface termination as investigated by all-electron B3LYP-D* simulations. *Theor. Chem. Acc.* **135:54** (2016)). On the other hand, the morphology of the synthetic grown crystals is highly dependent on the synthesis conditions such as pH, temperature and ion concentration. Platelet morphology grown along the *c*-axis can be attained in the presence of simulated body fluids (SBF) and such crystals can be used for prediction of in vivo bioactivity on implants (Muller F.A. et al., Preferred growth orientation of biomimetic apatite crystals. *J. Crystal Growth* **304**, 464 (2007)). The platelet morphology has been shown to be the result of low temperature (38 °C), increased pH values (in order to increase the growth rate) and has been associated with phosphate-rich environment (Kobayashi T. et al., Morphological variation of HAp grown in aqueous solution based on SBF. *Cryst. Eng. Comm.* **14**, 1143 (2012)), conditions similar to the synthesis route used in our study.

With regards to the anisotropy of interfacial energy, growth of apatite has been shown to attain a specific orientation within confined domains of collagen matrices and synthetic confinements as opposed to its growth in free media, which results in isotropic

and randomly oriented micron-sized crystals. This might be result of the templating effect of collagen (Wang Y. *et al.* The predominant role of collagen in the nucleation, growth, structure and orientation of bone apatite. *Nature Materials* **11**, 724 (2012) in combination with a higher cost of overcoming the interfacial energies for extrafibrillar mineralization. This was recently demonstrated in the recent work by Kim et al (Kim *et al.* The role of confined collagen geometry in decreasing nucleation energy barriers to intrafibrillar mineralization. *Nature Communications* **9:962**, 1-9 (2018)). The authors showed that the interfacial energies (α) for apatite nucleation and growth was specifically lower within the gaps of collagen fibrils in the presence of non-collagenous protein mimicking molecules as opposed to extrafibrillar mineralization. Moreover, the authors also show that the growth attained a single orientation inside the fibrils. Another recent work demonstrated that the growth of apatite in an isotropic synthetic matrix resulted in HAp crystals that were aggregated in random directions over the micron-scale, however when the HAp was mineralized within a unidirectionally stretched polymer matrix, the growth of HAp was anisotropic along the *c*-axis, which aligned with the stretching direction (Fukao, K. *et al.* Anisotropic Growth of Hydroxyapatite in Stretched Double Network Hydrogel. *ACS Nano* **11**, 12103-12110, (2017).

Point 6:

“Overall, I think that this work has some high quality data that deserves publication. However, in my opinion the analysis of the relevance to bone mineralization should undergo a major revision where the findings of this paper are put in context with all previous literature on bone mineralization and the role of collagen and other non-collagenous components.”

Authors comment:

We thank the reviewer for the encouraging remarks.

The text in the *Relevance to bone mineralization* has been revised to incorporate these suggestions, as described in Point 3 of Reviewer #2 with references to the following papers.

References:

1. Boskey, A.L. and A.S. Posner, Conversion of Amorphous Calcium Phosphate to Microcrystalline Hydroxyapatite - Ph-Dependent, Solution-Mediated, Solid-Solid Conversion. *Journal of Physical Chemistry*, 1973. 77(19): p. 2313-2317.
2. Harries, J.E., D.W.L. Hukins, C. Holt, and S.S. Hasnain, Conversion of Amorphous Calcium-Phosphate into Hydroxyapatite Investigated by Exafs Spectroscopy. *Journal of Crystal Growth*, 1987. 84(4): p. 563-570.

3. Tao, J.H., H.H. Pan, Y.W. Zeng, X.R. Xu, and R.K. Tang, Roles of amorphous calcium phosphate and biological additives in the assembly of hydroxyapatite nanoparticles. *Journal of Physical Chemistry B*, 2007. 111(47): p. 13410-13418.
4. Baht, G.S., G.K. Hunter, and H.A. Goldberg, Bone sialoprotein-collagen interaction promotes hydroxyapatite nucleation. *Matrix Biology*, 2008. 27(7): p. 600-608.
5. Boskey, A.L., Mineral-Matrix Interactions in Bone and Cartilage. *Clinical Orthopaedics and Related Research*, 1992(281): p. 244-274.
6. Nudelman, F., K. Pieterse, A. George, P.H.H. Bomans, H. Friedrich, L.J. Brylka, P.A.J. Hilbers, G. de With, and N.A.J.M. Sommerdijk, The role of collagen in bone apatite formation in the presence of hydroxyapatite nucleation inhibitors. *Nature Materials*, 2010. 9(12): p. 1004-1009.
7. Rhee, S.H., J.D. Lee, and J. Tanaka, Nucleation of hydroxyapatite crystal through chemical interaction with collagen. *Journal of the American Ceramic Society*, 2000. 83(11): p. 2890-2892.
8. Chen, Y., B.S. Bal, and J.P. Gorski, Calcium and Collagen Binding-Properties of Osteopontin, Bone Sialoprotein, and Bone Acidic Glycoprotein-75 from Bone. *Journal of Biological Chemistry*, 1992. 267(34): p. 24871-24878.
9. Brown, W.E., Crystal growth of bone mineral. *Clinical orthopaedics and related research*, 1966. 44: p. 205-220.
10. Bodier-Houille, P., P. Steuer, J.C. Voegel, and F.J.G. Cuisinier, First experimental evidence for human dentine crystal formation involving conversion of octacalcium phosphate to hydroxyapatite. *Acta Crystallographica Section D-Biological Crystallography*, 1998. 54: p. 1377-1381.

Reviewer #3

Point 1:

“The introduction suggests that we will get a real insight into what happens mechanistically in vivo, but then it turns out another in vitro study. Given the fact that in this this study a synthetic scaffold is used, the importance of the work is a level lower than the studies using collagen as the template (Gower, Beniash, Sommerdijk, Nassif, Tay). In addition, the TEM experiments are conventional (dry)TEM experiments – in contrast to the work of Sommerdijk, Nassif and Tay), so they provide very limited information even on the in vitro mineralization process. Essentially, the authors show the solid state transformation of ACP to apatite in a synthetic environment, in fact in vacuum (which again may affect the transformation). I therefore must conclude that the paper is not of sufficient interest for the broad audience of Nature Communications, and should better be submitted to a more specialized journal. Moreover, the authors make several poorly informed statements, which are in fact dangerous to the field if they are published like this. So I cannot support publication in this form.”

Authors comment:

Monitoring the morphological and structural transformation of ACP to apatite, *in vivo* is very difficult and this fact is depicted by the lack of existing literature evidence regarding the exact steps (and their detailed description) that happen during the transformation process. The subject of bone mineralisation has been and still is a matter of debate and an extensive description of the entire process is yet to be revealed. The past efforts of many research groups, as indicated by the reviewer, have focussed on understanding the specific mechanistic steps prior to and during the ACP formation in the gap zones of collagen and characteristic features of the end result, i.e. the final morphology, crystal symmetry, chemistry and orientation of the matured apatite crystals. In general, the steps from the deposition of ACP to its structural evolution into apatite, *i.e.* the fact that the spherical or irregular amorphous particles transforming into thin apatite platelets, is well described. However, there is a lack of clear understanding regarding the intermediate stages during the transformation that might be formulated as; (1) how does the initial crystalline nuclei form in the ACP particles? And (2) how does the crystalline domain grow and attain a uniform orientation going from a spherical or irregular morphology to a nanocrystalline platelet with high aspect ratio? Our work attempts to address these questions based on; (1) the morphological similarity of the initial ACP particles and (2) the morphological and structural similarity of the final apatite platelets, observed in our system to those observed during bone mineralisation.

We acknowledge the work of the groups referred to here by the reviewer (with References in our manuscript) and especially studies that demonstrate the role of ionic pre-nucleation clusters in the growth of ACP and its subsequent transformation. This theory, originally proposed by Posner and Betts (Ref [12] in our manuscript), has triggered the idea of cluster growth units, which has been the foundation of our work. The ACP-apatite transformation mechanism described in our study relies on the idea that material migrates to the growth front in clusters rather than adatoms, as in classical step flow, and the observed sizes match the previously published works. We acknowledge that this idea is not new (see Ref. [15-20] in our manuscript), however this is the first time that such a stepped growth mechanism is being described by using HRTEM in crystal sizes that match those of bone apatite rather than in cleaved surfaces of macroscopic hydroxyapatite single crystals.

We acknowledge that the synthetic scaffold used in our work is not the same as the collagen matrix in bone, however the similarity of the apatite crystals in terms of morphology, size and structure to bone apatite is striking. This is the basis for extending the

validity of our findings to the biological systems. Furthermore, as elaborated in the manuscript text, the 3D organization of the polymer matrix with ordered confined spaces plays an important role in the growth and orientation of the apatite crystals with properties mimicking bone-apatite. This point has been elaborated in the works by Nassif and co-workers (see Ref 31, Wang et. al. The predominant role of collagen in the nucleation, growth, structure and orientation of bone apatite *Nature Materials* **11**, 724 (2012)), where the authors emphasize on the importance of the 3D organization of collagen fibrils and the respective confined spaces for the growth and orientation of apatite.

Point 2:

“Examples: Line 34 – there is no hard evidence that ACP is delivered by intracellular vesicles.”

Authors comment:

This statement is derived from the References cited as 7-8-9 and 11 in our manuscript. Since, as mentioned in this Point, there is no hard evidence the text in Line 34 was re-phrased to describe it as a ‘*possible mechanism*’ taking place and the “*mineral containing vesicles*” term was also added.

Point 3:

“Line 40 – not all these papers refer to biological systems. Although all these papers are relevant, some are biomimetic systems.”

Authors comment:

We have clarified this paragraph in Line 34 by stating “*There is increasing in vivo and in vitro evidence indicating an amorphous calcium phosphate (ACP) precursor phase*”. Therefore, the References indicated in Line 40 also include biomimetic systems which incorporate the ACP as the apatite precursor phase.

Point 4:

“Line 41 – “while there is a continuous confirmation ...” I am not sure what is meant by this. Can the authors give the examples? I am not sure there is a continuous stream of evidence for this.”

Authors comment:

We stated that there is “*continuous confirmation...*” based on the recently published works on both *in-vivo* systems such as the growing bones of zebra fish fins (Mahamid J. et al., Amorphous calcium phosphate is a major component of the forming fin bones of zebrafish: Indications for an

amorphous precursor phase. *Proc. Natl Acad. Sci. USA* **105**, 12748–12753 (2008)), (Mahamid J. et al. Mapping amorphous calcium phosphate transformation into crystalline mineral from the cell to the bone in zebrafish fin rays. *Proc. Natl Acad. Sci. USA* **107**, 6316 (2010) and in mouse calvaria (Mahamid J. et al. Bone mineralization proceeds through intracellular calcium phosphate loaded vesicles: A cryo-electron microscopy study. *Journal of Structural Biology* **174**, 527 (2011)) and embryonic chicken bones (Kerschnitzki M. et al. Bone mineralization pathways during the rapid growth of embryonic chicken long bones, *Journal of Structural Biology* **195**, 82 (2016)) as well as multiple *in-vitro* studies that has demonstrated the presence of ACPs prior to the growth and crystallization to apatite within the gap-zones of collagen (Nudelman F. et al. The role of collagen in bone apatite formation in the presence of hydroxyapatite nucleation inhibitors. *Nature Materials* **9**, 1004 (2010) and Boonrungsiman S. et al. The role of intracellular calcium phosphate in osteoblast-mediated bone apatite formation. *PNAS* **109** (35), 14170 (2012)). For better expression of the sentence, we have rephrased the term “*continuous confirmation*” to “*with growing evidence for an ACP mediated growth of apatite in vivo and in vitro*” highlighted in Lines (33-40).

Point 5:

“Figure 6. It is a bit bold to state that this is “the current model for bone mineralization”. The community will agree there are many things unclear.”

Authors comment:

We agree with the reviewer that this model is not fully established. However, in light of recent work by Mahamid *et al.* (2010) (Ref. 7 in manuscript) and Boonrungsiman S. *et al.* (2016) (Ref. 9 in manuscript), we believe this could be a proposed model for the mineralization processes in bone. The Figure 6 caption has now been changed to “*Diagram describing a possible bone mineralization mechanism and how our work fits into the process*” in order to be presented as a more generalised process in which only the part relevant to our results are described in detail. Additionally, to the part (i) of the Figure 6 caption, we have added the following text “*intracellular matrix vesicles bud from osteoblasts and transport ACP precursors as irregular or spherical shaped granules to the gap zones of the collagen matrix (one of the most accepted theories)*”.

Point 6:

“And more technically: The authors suggest that “taking the images 2 min apart” may limit beam damage. This is not true, beam damage is mainly depending on the total electron dose.”

Authors comment:

The issue of the material’s beam sensitivity was raised from the previous reviewers as well, so we refer to our response on Point 6 of the Reviewer 1.

Reviewers' Comments:

Reviewer #1:

Remarks to the Author:

I am satisfied with the responses and the modifications introduced in the text in response to my previous concerns. I have also carefully examined the comments of the other reviewers and the changes introduced by the authors in response of those comments. It is not up to me to agree (here) with the concerns or responses but I would like to point out that the paper focuses in an intense field of debate right now and contributes with interesting points to the recently reignited discussion on early stages of mineralization that is been going on for 30-40 years . This is the reason that makes me think that this manuscript should be published and may generate some feedback from the audience. I do not understand why this communication should be moved to a more specific journal as suggested by other reviewer since papers on this topic have been recently published in this journal.

Luis M. Rodríguez-Lorenzo

Reviewer #2:

Remarks to the Author:

In my opinion the paper has undergone an in depth revision following the recommendations of the reviewers.

I do not have any other comment to add.

It could be published as is.

Reviewer #3:

Remarks to the Author:

I am very disappointed to see this manuscript re-submitted without any essential improvement on the concerns I raised. The response made by the authors could not convince me that their system is comparable with in vivo system, or even with hydrated in vitro system. Also the effect of vacuum and electron beam to the crystallization process was not seriously discussed.

1. For the relevance between their work and the in vivo system, the authors claim it is based on:

(1) The morphological similarity of the initial ACP particles

I could not understand what this would mean. As mentioned by the authors, the ACP particles are spherical or irregular shaped, which is expected for an amorphous phase. What morphological difference could exist for two kinds of ACP particles? The only interesting similarity might be the size of the particles (50-100 nm), which is however also common for ACP particles.

(2) The morphological and structural similarity of the final apatite platelets

I am not sure what "structural similarity" the authors are referring to here. The morphology of the apatite platelets do resemble those in bone according to the numbers "4 x 20 x 50 nm" at Line 282. It is however not shown how these numbers are measured, especially the thicknesses (what is the deviation here?). Furthermore, the discussion in this manuscript is not relevant to this specific size at all, and I do not see why this size would help the authors to correlate the mechanisms they observed to bone formation.

(3) The authors also mentioned "the importance of the 3D organization of collagen fibrils and the

respective confined spaces for the growth and orientation of apatite”

This seems to be suggesting that the synthetic scaffold they used is essential, and played similar role with collagen. Again, I am not sure what “importance” the authors are mentioning here. But no matter what it is, it is not shown by the experimental results of this manuscript at all. In Figure 5 the authors seem to suggest that main role of the synthetic scaffold (PCCL matrix) is confining the CaP crystal growth and inducing the platelet-like morphology, which is similar to collagen as they described at Line 415-424. It is noteworthy that the synthetic scaffold used by the authors is actually very different from collagen not only in chemical composition, but also in the intermolecular spacing (~10 nm in the PLLC matrix vs ~1.5 nm in type I collagen, see Xu et al., Structure analysis of collagen fibril at atomic-level resolution and its implications for intra-fibrillar transport in bone biomineralization, Phys. Chem. Chem. Phys., 2018). The authors compared the intermolecular spacing of the PLLC matrix to the length of the collagen gap zone (~40 nm) at Line 95-96, which is misleading. I could not imagine how 10 nm sized pores could induce CaP platelets that are 2 nm in thickness, especially considering that these pores could be expanded by even ACP particles into ~30 nm sized, as shown in Figure 5.

2. The authors also emphasized the importance of their observation of the growth steps, and correlated it with the Posner clusters. As they claimed: This theory, originally proposed by Posner and Betts (Ref [12] in our manuscript), has triggered the idea of cluster growth units, which has been the foundation of our work. The ACP-apatite transformation mechanism described in our study relies on the idea that material migrates to the growth front in clusters rather than adatoms, as in classical step flow, and the observed sizes match the previously published works. We acknowledge that this idea is not new (see Ref. [15-20] in our manuscript), however this is the first time that such a stepped growth mechanism is being described by using HRTEM in crystal sizes that match those of bone apatite rather than in cleaved surfaces of macroscopic hydroxyapatite single crystals.

The step growth observed by the authors was previously reported in HAp system by in-situ AFM study (Ref. 15-20), and is actually expected for different size of HAp. Confirming this phenomenon using HRTEM is not a ground-breaking result. One seemingly attractive statement made by the authors is that the steps are related to the Posner clusters, since the height of the steps matches with the size of Posner clusters. However, it has been well established in classical theories that the height of the growth steps should be related to the size of crystal unit cell, as also mentioned by the authors at Line 158. And I do not understand why it has to do with the Posner clusters. Actually, Posner clusters are usually visible in the ACP particles when visualized in cryoTEM under low dose (Ref. 6, 13, 14). Those clusters are however invisible in this study, suggesting their structures were already disrupted by the electron beam and vacuum. It is difficult to imagine how these clusters could then still exist and immigrate to the growth front, as proposed by the authors.

Another statement made by the authors at Line 298-300 is that “new layers start to form before the complete growth of the first layer, something that is not so consistent with the “classical” layer-by-layer growth mechanism in epitaxy.” Could the authors provide a reference for this statement? Do they suggest that in “classical layer-by-layer growth”, only one layer/step could be observed?

I feel the authors were only following the proposal made in Ref. 16 here, which suggests that the growth units for HAp crystal are Posner clusters since the growth kinetics of the steps are too slow. If this is the case, the authors should make clear that this is a proposal made by previous literatures, since there is no discussion on the growth kinetics of the steps in this study. Again, this questions the importance of the current study.

3. For the effects of electron beam and vacuum, which were also mentioned by the other 2 reviewers, the authors claim that:

(1) Answer to point 2, reviewer 2: in our experiments we did not observe the generation of any new crystalline material (or domains) within the amorphous matrix of the ACP particles as induced by the electron beam.

It is clearly shown in their Figure 2 that new crystalline material is forming. How could the authors confirm that this is not due to electron beam and/or vacuum? Actually, considering the aging time require to induce crystallization outside TEM is ~days, and the growth process observed in TEM is ~mins, it is very likely that most of the phenomena observed by the authors (e.g., nucleation on ACP particle surface, shrinking of the ACP particle, crystallization) are induced by the electron beam and/or vacuum. The only reliable observations might be the step growth, which is however expected, as I mentioned above.

(2) Line 492-493, The majority of the TEM data presented in this work was obtained from regions with little or no presence of the polymer.

I cannot understand how the authors could distinguish polymers from ACP, which are both amorphous and beam sensitive; and also how not showing the "regions with polymers" would benefit the discussion on the possible beam effect.

(3) Line 503-504: short exposure times were employed (0.5-0.8 s)

I find it strange that the authors insist to describe their imaging conditions using only exposure times. Could the authors provide any information about the electron dose they used?

In summary, I have to conclude that this manuscript still contains massive defects and is not ready for publication yet. The authors should carefully discuss the effect of vacuum and electron beam to the crystallization process, and build all of their statements on experimental results and references and in a more clear and solid way. Even after that, I am afraid this work would not be worthy for further consideration in Nature Communications.

Response to Referees' comments:

Reviewer #1 and #2

We kindly thank Reviewers #1 and #2 for their comments and for finding their issues and concerns adequately answered in the revised manuscript.

Reviewer #3

"I am very disappointed to see this manuscript re-submitted without any essential improvement on the concerns I raised. The response made by the authors could not convince me that their system is comparable with in vivo system, or even with hydrated in vitro system. Also the effect of vacuum and electron beam to the crystallization process was not seriously discussed."

Point 1:

"1. For the relevance between their work and the in vivo system, the authors claim it is based on: (1) The morphological similarity of the initial ACP particles. I could not understand what this would mean. As mentioned by the authors, the ACP particles are spherical or irregular shaped, which is expected for an amorphous phase. What morphological difference could exist for two kinds of ACP particles? The only interesting similarity might be the size of the particles (50-100 nm), which is however also common for ACP particles."

Authors comment:

We agree with the reviewer that only referring to the morphology of ACP particles is insufficient to validate our model. However, we have stated in the manuscript that the validation is not only based on the size and morphology of the ACP particles, but also on the size, chemistry and morphology of the transformed apatite platelets. Moreover, from our earlier studies, we have observed that when apatite platelets are formed within the confinement of the PLLC matrix, the initially formed ACP particles of 30-40 nm in size transforms *in-situ* to apatite platelets of 30-40 nm in length and 1.5-4 nm in thickness, which is in close similarity to sizes observed for bone-apatite. In contrast, when the same process was performed in an unconfined aqueous solution without the PLLC matrix, the ACP particles (with little variation in morphology) transformed into apatite platelets with a significantly larger variation in size with some platelets extending to lengths

greater than 200 nm (He et. al. Chem. Mater. 2012, 24, 892). Additionally, when ACPs transform to apatite within confinements, the chemistry of the apatite platelets has also been shown to match that of bone-apatite, for example: carbonation and crystallinity (He et. al. Chem. Mater. 2012, 24, 892).

An additional reason for claiming the similarity of ACP and apatite obtained in this study to that of bone-apatite is the disordered and hydrated layer surrounding the matured apatite platelets. A study by Wang *et. al.* suggests that such surface layer provides an interface through which water molecules are able to orient the apatite crystals *in vitro*, without the presence of any biological component (Wang Y. *et al.* Water-mediated structuring of bone apatite. *Nature Materials* **12** (12), 1144 (2013)). In our experiments such a disordered surface layer has been identified at all the aging stages (water induced).

Therefore, the validation of our model is not only based on one aspect of morphological similarity i.e. ACP, but a combination that includes both the ACP and apatite particle size, structure, chemistry and orientation.

Point 2:

“(2) The morphological and structural similarity of the final apatite platelets. I am not sure what “structural similarity” the authors are referring to here. The morphology of the apatite platelets do resemble those in bone according to the numbers “4 x 20 x 50 nm” at Line 282. It is however not shown how these numbers are measured, especially the thicknesses (what is the deviation here?). Furthermore, the discussion in this manuscript is not relevant to this specific size at all, and I do not see why this size would help the authors to correlate the mechanisms they observed to bone formation.”

Authors comment:

With regard to the measured size of the platelets, the thickness was indeed the most accurately measured dimension and was derived from several HRTEM images of different crystallites. The deviation in the measured size was 0.5 nm and that deviation also includes differences in thickness due to the step traces on the sides (see Figure 4 a and b in the manuscript). The biggest deviation though, comes from the wide facets of the crystals due to their imperfect shape, as shown in Figure 4c, which demonstrates that the average width corresponds to the maximum observed width.

In the interest of our discussion, the irregular shape of the platelets is equally important (given the striking similarity to natural bone-apatite crystallites). We believe that the proposed growth mechanism is an explanation for the platelet shape obtained in our

system, which might be extended to be a possible mechanism for the growth of natural apatite. Apatite crystals left to grow in an unconfined environment and under thermodynamic equilibrium, do not exhibit such small sizes; they can grow in prisms of several microns in length with apparent faceting. So, the size and the shape/morphology help us to correlate the mechanisms observed in our study to the natural system. In effect, this is what we refer to as the structural similarity throughout the study.

Point 3:

“(3) The authors also mentioned “the importance of the 3D organization of collagen fibrils and the respective confined spaces for the growth and orientation of apatite”.

This seems to be suggesting that the synthetic scaffold they used is essential, and played similar role with collagen. Again, I am not sure what “importance” the authors are mentioning here. But no matter what it is, it is not shown by the experimental results of this manuscript at all. In Figure 5 the authors seem to suggest that main role of the synthetic scaffold (PCCL matrix) is confining the CaP crystal growth and inducing the platelet-like morphology, which is similar to collagen as they described at Line 415-424. It is noteworthy that the synthetic scaffold used by the authors is actually very different from collagen not only in chemical composition, but also in the intermolecular spacing (~10 nm in the PLLC matrix vs ~1.5 nm in type I collagen, see Xu et al., Structure analysis of collagen fibril at atomic-level resolution and its implications for intra-fibrillar transport in bone biomineralization, Phys. Chem. Chem. Phys., 2018). The authors compared the intermolecular spacing of the PLLC matrix to the length of the collagen gap zone (~40 nm) at Line 95-96, which is misleading. I could not imagine how 10 nm sized pores could induce CaP platelets that are 2 nm in thickness, especially considering that these pores could be expanded by even ACP particles into ~30 nm sized, as shown in Figure 5.”

Authors comment:

We certainly understand and have acknowledged in the manuscript the differences between our polymeric system versus collagen in bone. However, we believe that the PLLC matrix provides as a valid model for collagen because of its ability to provide a three-dimensional arrangement of nanoscale confinements for the mineralization of the calcium phosphate species, which we have elaborated in previous studies and can be summarized as follows; (1) **Selectivity**: Mineralization of CaP within the PLLC confinements has shown to govern the type of CaP formed. In one of our earlier papers (Rajasekharan et. al. Cryst. Growth Des. 2015, 15, 2775) we have demonstrated that when CaPs form within the nanoscale confinements greater than 10 nm, a mixture of bone-like apatite and acidic polymorphs form and it was necessary to maintain the confinements (or pores) of the PLLC to less than 10 nm to selectively and homogeneously form bone-like

apatite platelets. (2) **Size and chemistry:** We have observed that when apatite platelets are formed within the confinements of the PLLC matrix, they form initial ACP particles of 30-40 nm in size, which transform *in-situ* to apatite platelets of 30-40 nm in length and 1.5-4 nm in thickness, in close similarity to sizes observed for bone-apatite. In contrast, when the same process was performed in an unconfined aqueous solution without the PLLC matrix, the apatite platelets had high variation in size with some platelets extending to lengths greater than 200 nm (He et. al. Chem. Mater. 2012, 24, 892). Additionally, the chemistry of the apatite platelets has also been shown to match that of bone-apatite, for example: carbonation and crystallinity (He et. al. Chem. Mater. 2012, 24, 892). (3) **Orientation:** The anisotropic geometry of the PLLC confinements i.e. pores surrounded by hexagonally ordered fibrils, is able to orient the apatite particles along the *c*-axis and over the long-range. This specific feature of orienting the apatite crystals within confined domains has been demonstrated in He et. al. Adv. Mater. 2015, 27, 2260 and Rajasekharan et. al. Small, 2017, 13, 1700550. Furthermore, the aforementioned importance of mineralizing CaPs within confined nanoscale domains has also been demonstrated in other synthetic systems such as track etched alumina membranes where the authors have also shown that nanoscale confinements, up to 100 nm in size, can have a significant effect on the selectivity, size and orientation of apatite platelets (Cantaert et. al. J. Mater. Chem. B 2013, 1, 6586, Wang et. al. Chem. Mater. 2014, 26 5830 and Cantaert et. al. Chem.-Eur. J. 2013, 19, 14918). Recent *in-vitro* studies on the role of collagen, in the absence of secondary molecules such as non-collagenous proteins, during mineralization has also pointed that a highly dense and 3D supra-fibrillar assembly of the collagen fibrils is sufficient for the growth and orientation of the apatite crystals with chemical and morphological properties mimicking bone-apatite (see Ref 31, Wang et. al. The predominant role of collagen in the nucleation, growth, structure and orientation of bone apatite *Nature Materials* **11**, 724 (2012)). The study also supported that the structure of collagen had a direct effect on the mineralization processes.

While we surely agree with the Referee that the PLLC matrix is chemically different from the collagen matrix, the aforementioned experimental evidence that nanoscale confinements and its 3D geometry of the surrounding matrix can selectively form apatite platelets that are morphologically and chemically similar to bone-apatite is our main motivation for extending our PLLC model to collagen fibrils. Furthermore, the experimental evidence supporting (from this study and previous studies) the similarity in chemistry and morphology the initial (ACP) and final (apatite) CaP species forming within the PLLC is worthy of consideration.

The Referee also brings up an interesting point with respect to the difference in interfibrillar spacing between the PLLC matrix and collagen matrix and its subsequent relation to crystal thickness. The interfibrillar spacing of the PLLC matrix is approximately 10 nm, where the CaP species are located and nucleate into ACP particles, further transforming into apatite. In collagen, the initial mineral (ACP) deposition is at the 40 nm long gap zones of the collagen fibrils, from where the growth and maturation of apatite platelets begin, followed by penetration into the fibril spaces of 1.5 nm. The similarity in localization of initial mineral deposition and transformation acts as the basis for comparing the PLLC and collagen systems in our work. Although, it is possible that the 1.5 nm spaces between collagen fibrils might contribute for a 2 nm thickness of bone-apatite, it is worthwhile to note that the actual thickness of bone-apatite crystals falls within a range of 2-6 nm (Dunlop J.W.C & Fratzl P, *Annual Review of Materials Science* **40**, 1 (2010) and Reznikov et al., *Science* 360, eaao2189 (2018)). This poses an open question, what really contributes towards maintaining the thickness of the apatite crystals within confinements, be it collagen *in-vivo* or collagen (or synthetic matrices like PLLC) *in-vitro*? Is it solely the confinement that forces the thickness or does the confinement acts as a trigger to direct the crystal to maintain within a specific range of thicknesses, in other words a thermodynamic property of apatite? In the current study, we believe that the PLLC system is able to select for ACP, constrains its size via elastic deformation and further orients the growth of apatite along its *c*-axis within the 10 nm confinements. However, the final platelet thickness i.e. 4 nm, might be a related effect from the confinement as well as the thermodynamics of the crystal growth. We acknowledge that these questions need to be evaluated further.

In summary, the current study provides a model for understanding a specific aspect of the bone mineralization with the consideration of experimentally verified factors as elaborated above. Although our system has differences compared to bone and specifically collagen, the mechanism of transformation from ACP to apatite observed here might provide as a guidance for the further exploration of the transformation mechanism *in-vivo*. The results from our work also indicate that, the strongly contested theory of an ACP precursor for bone-apatite might possibly be valid with respect to our proposed mechanism of transformation.

Point 4:

“2. The authors also emphasized the importance of their observation of the growth steps, and correlated it with the Posner clusters. As they claimed: This theory, originally proposed by Posner and Betts (Ref [12] in our manuscript), has triggered the idea of cluster growth units,

which has been the foundation of our work. The ACP-apatite transformation mechanism described in our study relies on the idea that material migrates to the growth front in clusters rather than adatoms, as in classical step flow, and the observed sizes match the previously published works. We acknowledge that this idea is not new (see Ref. [15-20] in our manuscript), however this is the first time that such a stepped growth mechanism is being described by using HRTEM in crystal sizes that match those of bone apatite rather than in cleaved surfaces of macroscopic hydroxyapatite single crystals. The step growth observed by the authors was previously reported in HAp system by in-situ AFM study (Ref. 15-20), and is actually expected for different size of HAp. Confirming this phenomenon using HRTEM is not a ground-breaking result.”

Authors comment:

We agree with Referee #3 that a stepped growth is expected for various sizes of HAp crystals, when the crystal already exists. We believe there is a fundamental difference between the systems described in Ref. 15-20 and our system, without suggesting that information of significant importance cannot be obtained from either. In the above references the growth starts from already cleaved and exposed surfaces and in many of the cases, steps and dissolution related spiral surface depressions pre-exist thereby providing “easy” and expected growth points. However, that is completely different from observing how apatite starts to nucleate from within an amorphous material and how the subsequent growth occurs from the very beginning. While trying to understand how the bone apatite crystals obtain their irregular morphology and what happens during its structural transformation, we believe that a system, where the end product is much more similar in terms of crystallinity and morphology, is more appropriate than prisms of several microns in size that have never been observed in bone (as described in ref. 15-20). From our point of view, this work not only simply confirms a well-established mechanism; but also demonstrates that the mechanism is present from the start of nucleation from the amorphous state, which to our knowledge, has never been described using a high-resolution imaging technique. We anticipate that our results might provide valuable insight on bone mineralisation, which is still under investigation and continuous debate.

Point 5:

“One seemingly attractive statement made by the authors is that the steps are related to the Posner clusters, since the height of the steps matches with the size of Posner clusters. However, it has been well established in classical theories that the height of the growth steps should be related to the size of crystal unit cell, as also mentioned by the authors at Line 158. And I do not understand why it has to do with the Posner clusters. Actually, Posner clusters

are usually visible in the ACP particles when visualized in cryoTEM under low dose (Ref. 6, 13, 14). Those clusters are however invisible in this study, suggesting their structures were already disrupted by the electron beam and vacuum. It is difficult to imagine how these clusters could then still exist and immigrate to the growth front, as proposed by the authors.”

Authors comment:

We would like to bring to notice a fundamental difference between the given references and our work; we initiate our observations after the formation of ACP. Our focus is on the transformation from the amorphous spherical state to unidirectionally orientated apatite platelets. This is post ACP formation, unlike the pre-nucleation stages described in the referenced works. The imaging of the Posner’s clusters or any other pre-nucleation clusters is not in the scope of this specific study. Additionally, since the ACP formation is taking place outside the TEM, the cluster structures are not necessarily disrupted by the electron beam.

Our observations also demonstrate that samples which have been aged for around 1 week, exhibit a variation in the growth steps, i.e., from amorphous particles to partially crystallised to fully transformed platelets. As we state in our manuscript in Lines 126-128: “...signifying that the aging procedure occurs at varying rates throughout the whole sample. At the same time, this difference in growth rate allows the observation of different aging steps in the sample”. While we do not expect to observe any pre-ACP nucleation state of the sample after 1-week aging, we have occasionally observed networks of small nm sized clusters as shown in Figure 1 in this letter, which is similar to what has been previously observed by Habraken *et al.* in Ref. 14 of our manuscript. This result has not been included in the manuscript since the focus was not on the ACP formation, rather on what happens after the formation and subsequent crystallization. Moreover, such results have already been shown in the aforementioned reference. Therefore, the similarity in the structure together with results from previous literature makes us confident that the nm clusters could be present at the pre-ACP stage and might also be a building block of HAp.

Furthermore, while the size of the steps (i.e. step heights) is similar to the interplanar spacing of the {01-10} planes of HAp, it is important to note that this similarity is only observed (specifically) along that direction and not along other directions. We have observed the step formation from multiple zone axes and all the observations indicate that the average area included in the steps match to the previously observed sizes of the pre-nucleation (or Posner’s) clusters reported in the literature.

Figure 1: Network formation of nm sized particles observed in samples aged for 1 week.

Point 6:

“Another statement made by the authors at Line 298-300 is that “new layers start to form before the complete growth of the first layer, something that is not so consistent with the “classical” layer-by-layer growth mechanism in epitaxy.” Could the authors provide a reference for this statement? Do they suggest that in “classical layer-by-layer growth”, only one layer/step could be observed?”

Authors comment:

In classical “layer-by-layer” growth mechanism in epitaxy or else Frank-van Der Merwe (FM) growth mechanism, which was established in 1949 (One-dimensional dislocations. II. Misfitting Monolayers and Oriented Overgrowth. F.C. Frank and J.H. van der Merwe. Proceedings of the Royal Society of London, Series A, Mathematical and Physical Sciences, Vol. 198, No 1053, pp. 216-225), multiple steps can exist on the same layer (surface sites) and adatoms preferentially attach to the surface site in order to form a smooth layer. This is a 2D growth mode and a complete layer has to be formed before the formation of a new layer. In that sense, our observations of multiple new layers at the same time, differentiates our growth mechanism and also does not support the presence of adatoms as growth units.

Point 7:

“I feel the authors were only following the proposal made in Ref. 16 here, which suggests that the growth units for HAp crystal are Posner clusters since the growth kinetics of the steps are too slow. If this is the case, the authors should make clear that this is a proposal made by previous literatures, since there is no discussion on the growth kinetics of the steps in this study. Again, this questions the importance of the current study.”

Authors comment:

We agree that our proposal regarding the claim that Posner’s clusters are growth units for apatite is based on previous works, which also includes the original work by Posner and Betts (Posner A.S. & Betts F. Synthetic amorphous calcium phosphate and its relation to bone mineral structure. *Acc. Chem. Res.* **8**(8), 273 (1975)). We believe that, given the presence of Posner’s clusters in the final apatite structure, it is possible that the clusters do play an important role during the transformation of ACP to apatite. We acknowledge that this is an open question and requires further understanding on the kinetics of the transformation and the exact manner in which the clusters take part in the process. As for the importance of our current study, we believe it is the actual mode of transformation, *i.e.* the step-wise growth from within the ACP matrix into the unidirectionally oriented apatite platelet within the realms of bone-apatite chemistry and structure. This study, as mentioned earlier as well, provides a benefit in that it sheds light on an as yet unknown mechanism of bone mineralization but also encourages deeper probing into more fundamental questions relating to the kinetics and mode of transformation *in-vivo*, which cannot be undermined.

Point 8:

“3. For the effects of electron beam and vacuum, which were also mentioned by the other 2 reviewers, the authors claim that:

(1) Answer to point 2, reviewer 2: in our experiments we did not observe the generation of any new crystalline material (or domains) within the amorphous matrix of the ACP particles as induced by the electron beam.

It is clearly shown in their Figure 2 that new crystalline material is forming. How could the authors confirm that this is not due to electron beam and/or vacuum? Actually, considering the aging time require to induce crystallization outside TEM is ~days, and the growth process observed in TEM is ~mins, it is very likely that most of the phenomena observed by the authors (e.g., nucleation on ACP particle surface, shrinking of the ACP particle, crystallization) are induced by the electron beam and/or vacuum. The only reliable observations might be the step growth, which is however expected, as I mentioned above.”

Authors comment:

In our response to the point 2 of the 2nd Reviewer we started by stating that *“While it is possible that the electron beam provides the energy for atomic mobility that might promote the extension of a pre-existing crystalline region, in our experiments ...”*. It is very clear that we observe extension of pre-existing crystalline areas and that is very possible to be an effect of the beam, however we do not observe any new nucleation of material. The area 2, indicated in Figure 2(c), was already visible from the start and is indicated by the lower right magenta arrow in Figure 2 (a), along with several other areas within the particle; the arrows indicate the most evident ones.

We understand the concern regarding the aging time and that might appear as a rapid transformation but the samples in which we observed the stepped growth (Figure 2 and 3) have been aged for at least 1 week outside the TEM. Most of the ACP particles have been transformed to a semi-polycrystalline state prior to the observation. As we mentioned above and our previous comments to the other reviewers, the TEM beam can provide the required energy for further growth promotion. However, it doesn't mean that the growth or transformation could have proceeded in a very different pathway considering the structural complexity of the apatite crystals.

Regarding the step growth being an expected result please refer to our comment in **Point 4**.

Point 9:

“(2) Line 492-493, The majority of the TEM data presented in this work was obtained from regions with little or no presence of the polymer.

I cannot understand how the authors could distinguish polymers from ACP, which are both amorphous and beam sensitive; and also how not showing the “regions with polymers” would benefit the discussion on the possible beam effect.”

Authors comment:

While both the polymer and the ACP are amorphous materials, our system can be described as a composite material, where spherical amorphous particles are incorporated within a polymer matrix. The fact that both are amorphous doesn't change the fact that the two constituents do not exhibit the same density and moreover the ACP initial spherical morphology is very distinct (in addition to the contrast enhancement that it offers). Figure 2 (a) shows that even in SEM the ACP particles are present and can be easily distinguished from the polymer fibrils.

The areas of low mineral content, as in the case of freshly prepared samples in which the majority of CaP is still amorphous with a spherical geometry or as small crystallites (in the range of 4-5 nm as in Figure 3), have been proven to be more sensitive under the beam. To be able to observe these nanosized crystals, a very thin area and high magnification is needed. The presence of polymer not only disturbs the imaging (especially when surrounding the particles) but also is much more sensitive to the electron beam than the ACP. This would lead to the areas of interest being potentially destroyed from beam heating, especially at higher magnifications (over x600k, where the electron dose can be increased significantly). In HRTEM mode, the imaging has been proven to be challenging when high amount of polymer is present and we found it more suitable to remove the polymer prior to imaging or at least limit the imaging to the regions without the polymer. We have earlier added this information in the Experimental Section as: *“The TEM specimens also contained the PLLC matrix and some areas having larger fractions of polymer were quite unstable and not easy to image due to beam-induced heat. For this reason, we have not included TEM observations of such areas. However, in some cases (as seen in Fig. 3) remnants of the polymer were present near the apatite crystals outside of the immediate viewing area, which still induced some drift and charging effects of the whole region”*. This is relevant to the sample preparation as well as for the imaging conditions. Regarding the beam effect, noting that we remove the beam-sensitive polymer means that we also lower possible radiolysis due to heating.

We also must note here that samples of the same aging stages (even when consisting only of ACP) are more resistant to beam damage. Moreover, samples that contain high mineral concentration (as in samples with fully aged crystals) are even less sensitive, even in the presence of the polymer matrix. We have studied TEM lamella lift-outs of the composite to study the 3D confinement of the apatite crystals in our system and we have imaged crystals on the surface of the lamella in HRTEM without significant disturbance from the polymer.

Point 10:

“(3) Line 503-504: short exposure times were employed (0.5-0.8 s).

I find it strange that the authors insist to describe their imaging conditions using only exposure times. Could the authors provide any information about the electron dose they used?”

Authors comment:

Information regarding the electron dose were not included in the manuscript since the exact doses were not recorded during the initial experiments. We understand the importance of such information when dealing with beam sensitive materials and the concern of Referee #3 is justified. For that reason, additional TEM experiments were performed in which we also monitored the effect of the beam on the samples under continuous illumination for a period of 2-4 mins in similar magnifications. The electron dosage varied between 15600 and 8.86×10^5 e/nm²s depending on the magnification (x87k – x620k). As we stated also in the Experimental, in the case of fully aged crystals, no damage was observed and the crystals appeared unchanged and stable under the electron beam. In the case of ACP or partially crystallised particles, beam damage was observed after 2 mins for electron doses of $5-8 \times 10^5$ e/nm²s under continuous illumination. Therefore, the precautions described in the paragraph of page 24 (Lines 500-508) can significantly reduce the damaging effect of the beam and are common techniques used for sensitive materials, and we have been careful not to claim that the effects of the beam were completely diminished. Relevant information regarding the electron dosage were added in the paragraph in Lines 494-508.

Point 11:

“In summary, I have to conclude that this manuscript still contains massive defects and is not ready for publication yet. The authors should carefully discuss the effect of vacuum and electron beam to the crystallization process, and build all of their statements on experimental results and references and in a more clear and solid way. Even after that, I am afraid this work would not be worthy for further consideration in Nature Communications.”

Authors comment:

We hope and anticipate that our comments and responses on the various points raised by Referee #3 have been answered in a satisfactory level in this revised version.

Reviewers' Comments:

Reviewer #1:

Remarks to the Author:

Comments from reviewer 3.

I am very disappointed to see this manuscript re-submitted without any essential improvement on the concerns I raised. The response made by the authors could not convince me that their system is comparable with in vivo system, or even with hydrated in vitro system. Also the effect of vacuum and electron beam to the crystallization process was not seriously discussed.

Point 1. I could not understand the concern raised in this point by reviewer 3. Yes there is a similarity between the proposed model and the in vivo system. As the authors state, similarity is not only based on the size and morphology of the ACP particles, but also on the size, chemistry and morphology of the transformed apatite platelets. The presence of the hydrated layer is also intrinsic of an apatite phase. The model is valid as a starting point to do the study. Also no arguments are provided to think this is not a valid model.

Point 2: Before entering in the point raised, in the worst case scenario the information would be irrelevant but not contradictory with the proposed mechanisms. Then, Authors provide a response on how the size is measured. I agree that the expression "structural similarity" is confusing. It would be better to replace it with morphological similarity. The authors then introduce the point of the "irregular shape of the platelets" to defend the similarity between their model and biological apatites. There is no contradiction here either. Authors arguments are sound.

Point 3: This is an interesting discussion point. Is the PLLC spacing going to produce a similar platelet formation from ACP as collagen?: For reviewer 3, the confinement space in the PLLC is obviously different from the confinement space in collagen 10 vs 1.5nm. Authors claim in the rebuttal letter that the confinement space in collagen is not the interfibrillar collagen zone but the initial gap zones of the collagen fibrils that appears to be about 40 nm long. The results published previously by the authors of this manuscript show that this is the case as discussed. Thus, the model is valid though the differences about the confinement may deserve a further comment in the manuscript. Apatite start crystallizing along C axis anyway providing initially elongated shapes.

Point 4: Confirming a stepped growth mechanism for the crystallization of apatites is not ground-breaking as the reviewer says, still, it is important to accumulate evidences in a field that is open for discussion and evidences from HTEM have not been obtained before.

Point 5: Authors show the presence of clusters in the rebuttal letter and more importantly provide an explanation about why the clusters are not disrupted by the electron beam. The image should be added as supplementary materials and also figures 3 and 4 containing TEM images should add the aging time of the particles.

Point 6: The layer by layer classical theory is a 2D model. There is no contradiction with the differences in a more complex 3D model.

Point 7: I agree that authors are not proposing a new theory and the appropriate references should be included. About the importance of the paper is not in this point but rather in the observation by HTEM for the first time...

Point 8: This the key point of the paper. Whether, the observed crystal formation/transformations are affected by the beam. In the rebuttal letter, authors claim no observation of new crystallization points but accept that energy from the beam may have induce further growth promotion but, not affecting the proposed mechanism of growing. Thus authors are accepting the concern of reviewer three. Since the actual key point is whether the proposed mechanism is acceptably based on

evidence, I conclude that no concerns have been raised in this point.

Point 9: polymer and ACP domains can be distinguished as it is possible to appreciate in figures 1 and 2.

Point 10: The information required about the dosage has been incorporated.

Point 11: I cannot appreciate those massive defects. The main concern on this paper is whether ACP/recently crystallized apatite may be affected by the beam. Supplementary information has been incorporated to disregard this concern. Then, the second point is whether this work deserves to be published in Nature communications. This is the editor to decide but in my opinion the manuscript does not present a new ground breaking theory but it does presents a new (and not easy to obtain) HTEM evidence in a topic open for discussion.

Reviewer #3:

Remarks to the Author:

The authors argue in great length why the referees comments are not valid, but have not made significant changes to the manuscript.

I therefor see no reason to change my opinion on the quality of the manuscript and the novelty of the research described, and still recommend rejection.

Response to Referees' comments:

Reviewer #1

“Comments from reviewer 3. I am very disappointed to see this manuscript re-submitted without any essential improvement on the concerns I raised. The response made by the authors could not convince me that their system is comparable with in vivo system, or even with hydrated in vitro system. Also the effect of vacuum and electron beam to the crystallization process was not seriously discussed. Point 1. I could not understand the concern raised in this point by reviewer 3. Yes there is a similarity between the proposed model and the in vivo system. As the authors state, similarity is not only based on the size and morphology of the ACP particles, but also on the size, chemistry and morphology of the transformed apatite platelets. The presence of the hydrated layer is also intrinsic of an apatite phase. The model is valid as a starting point to do the study. Also no arguments are provided to think this is not a valid model.”

Point 1:

“Point 2: Before entering in the point raised, in the worst case scenario the information would be irrelevant but not contradictory with the proposed mechanisms. Then, Authors provide a response on how the size is measured. I agree that the expression “structural similarity” is confusing. It would be better to replace it with morphological similarity. The authors then introduce the point of the “irregular shape of the platelets” to defend the similarity between their model and biological apatites. There is no contradiction here either. Authors arguments are sound.”

Authors comment:

We agree how the term “structural similarity” could be confusing when referring only to the size and shape of the formed crystals. However, the phrase used in our manuscript is “structural and morphological” (Lines 105-106) and here we refer to both structural level (amorphous to crystalline) and morphological level (spheres to platelets of different sizes) (Lines 106-108). We also refer to a similar point in Lines 409-410 in our manuscript, where we refer to the final apatite crystal resemblance, we write: “(ii) the final apatite crystal resembles bone apatite in size, crystallinity, composition and irregular platelet-like geometry.” Size, crystallinity, composition and crystal geometry (shape) consist the morphological and structural similarity. In both cases both the structure and the morphology of the crystals are concerned.

At the same time, as per the reviewer's recommendation, we have removed or replaced the word "structural" in the following parts of the manuscript such that the word is not incorrectly used, to avoid any further confusion:

- Line 14: Changed the "structural transformation" to "transformation".
- Line 360: Changed the "structural model" to "morphological model".
- Line 455: Changed the "structural imperfectness" to "structural and morphological imperfectness".

Point 2:

"Point 3: This is an interesting discussion point. Is the PLLC spacing going to produce a similar platelet formation from ACP as collagen?: For reviewer 3, the confinement space in the PLLC is obviously different from the confinement space in collagen 10 vs 1.5nm. Authors claim in the rebuttal letter that the confinement space in collagen is not the interfibrillar collagen zone but the initial gap zones of the collagen fibrils that appears to be about 40 nm long. The results published previously by the authors of this manuscript show that this is the case as discussed. Thus, the model is valid though the differences about the confinement may deserve a further comment in the manuscript. Apatite start crystallizing along C axis anyway providing initially elongated shapes."

Authors comment:

We agree with the reviewer that commenting on the essential differences in confinement between the PLLC matrix and collagen is essential, which would further clarify the importance of the confinements in the PLLC matrix to obtain bone-like apatite and its corresponding relation to the collagen matrix in-vivo. And indeed, as stated in the previous rebuttal, the PLLC matrix and its nanoscale confinements are crucial for obtaining an apatite platelet formation as observed during collagen mineralization.

As per the reviewer's recommendation, we have added the following text in our manuscript with appropriate citations that comments on the relations between the confinements of the PLLC and collagen matrices, Line 414: *"It is important to note that the interfibrillar spacing of the PLLC matrix is approximately 10 nm, where the CaP species nucleate into ACP particles, further transforming into apatite. In collagen, the initial mineral (ACP) deposition is at the 40 nm long gap zones of the collagen fibrils, followed by the growth and maturation of apatite platelets, leading to its penetration into the fibril spaces of 1.5 nm. Even though the final fibril spacing is different between the PLLC and collagen, it must be emphasized that it is the similarity in localization of the initial mineral deposition, (i.e. ACP and its subsequent transformation) that is the primary basis for comparing the*

PLLCC of this work and collagen systems in-vivo. Although, it is possible that the 1.5 nm spaces between collagen fibrils might contribute for a 2 nm thickness of bone-apatite, it is notable that the actual thickness of bone-apatite crystals falls within a range of 2-6 nm. In this work, the thickness of the apatite platelets formed within the PLLC also falls in a similar range (≈ 4 nm). Therefore, the PLLC matrix provides a confined domain of a specific size, which is essential to obtain a final apatite particle similar to those observed in bone. Moreover, we have previously shown that the size of the PLLC confinements plays an important role in controlling the type of CaP formed. Confinements exceeding 10 nm results in a mixture of CaP phases while under 10 nm, the CaP particles purely resembled bone-like apatite platelets. Moreover, the confinement also restricts the growth of the crystallites in the range of 30-40 nm in length, while in absence of the PLLC matrix confinements, the crystallites could exceed 200 nm in length. Therefore, it is clear that the 10 nm confinements of our PLLC system is able to select for ACP, constrains the size via elastic deformation and favours the growth of apatite crystallites with thickness similar to bone-apatite and with their c-axis aligned along the length of the micellar fibrils. ”

Point 3:

“Point 4: Confirming a stepped growth mechanism for the crystallization of apatites is not ground-breaking as the reviewer says, still, it is important to accumulate evidences in a field that is open for discussion and evidences from HTEM have not been obtained before. Point 5: Authors show the presence of clusters in the rebuttal letter and more importantly provide an explanation about why the clusters are not disrupted by the electron beam. The image should be added as supplementary materials and also figures 3 and 4 containing TEM images should add the aging time of the particles.”

Authors comment:

The image of the pre-nucleation clusters has been added in the supplementary material in our manuscript as Extended Data Figure 1 and relevant text has been add to the main text in Line 134. Therefore the sentence in Lines 132-134 has changed to “*The clusters were thought to correspond to the first crystal units that aid the transformation to apatite via a rearrangement of CaP building blocks, similar to those observed by Habraken et al., since we observed similar arrangements in our samples (Extended Data Fig. 1).*”

The aging times of the particles depicted in Figures 3 and 4 are now added in the caption of the respective Figures.

Due to the addition of the Extended Data Figure 1, the numbering on the following Figures is changed in both the text and the respective captions.

Point 4:

“Point 6: The layer by layer classical theory is a 2D model. There is no contradiction with the differences in a more complex 3D model. Point 7: I agree that authors are not proposing a new theory and the appropriate references should be included. About the importance of the paper is not in this point but rather in the observation by HTEM for the first time...”

Authors comments:

Indeed, we anticipate that our work presents results that have been proposed in theory and presented experimentally with other techniques. As we also stated in our previous rebuttal letter, the observation of evidence for this theory by HRTEM for the first time would assist the advancement of the field by offering a further insights on a subject still open for discussion and a matter of debate for years; the results in our work correspond to a more relevant crystal size of apatite to the bone. Therefore, HRTEM is the suitable technique with the appropriate resolution to detect such structural transformations in the nanoscale; changes, which other techniques used in the literature to study this transformation (e.g. AFM), cannot detect.

The appropriate references regarding previously reported stepped growth correspond to the Ref. 15-20 in our manuscript. In the **Introduction Section** of our manuscript these works are being described as well as in other places in the manuscript:

- Line 286: Ref 19 is used to show that a dissolution-reprecipitation process is taking place simultaneously with the stepped growth, observed both in their work and ours.
- Line 310: Ref 15 is used to compare the size of the steps to the size of the pre-nucleation clusters observed in the literature.
- In Lines 384-387, we correlate our observations with the existing literature that discuss stepped growth in relation to cluster migration, of similar sizes as “Posner’s cluster” (Ref 15-17).

Point 5:

“Point 8: This the key point of the paper. Whether, the observed crystal formation/transformations are affected by the beam. In the rebuttal letter, authors claim no observation of new crystallization points but accept that energy from the beam may have induce further growth promotion but, not affecting the proposed mechanism of growing.

Thus authors are accepting the concern of reviewer three. Since the actual key point is whether the proposed mechanism is acceptably based on evidence, I conclude that no concerns have been raised in this point. Point 9: polymer and ACP domains can be distinguished as it is possible to appreciate in figures 1 and 2. Point 10: The information required about the dosage has been incorporated. Point 11: I cannot appreciate those massive defects. The main concern on this paper is whether ACP/recently crystallized apatite may be affected by the beam. Supplementary information has been incorporated to disregard this concern. Then, the second point is whether this work deserves to be published in Nature communications. This is the editor to decide but in my opinion the manuscript does not present a new ground breaking theory but it does presents a new (and not easy to obtain) HTEM evidence in a topic open for discussion.”

Authors comments:

There are no concerns raised in this part of the Reviewer’s report. We kindly thank the Reviewer for the valuable comments and feedback.

Reviewer #3

Point 1:

“The authors argue in great length why the referees’ comments are not valid, but have not made significant changes to the manuscript. I therefor see no reason to change my opinion on the quality of the manuscript and the novelty of the research described, and still recommend rejection.”

Authors comments:

In the review process of this manuscript we acknowledged and considered all three reviewers’ comments and suggestions equally. Through this process we attempted to further answer, clarify, support and strengthen our present work, rather than dismiss or lower the validity of any comment. We anticipate that the manuscript has been revised and changed appropriately as per the referees’ suggestions.

Best regards,
Dr. Martin Andersson